# LOW-RANK FEW-SHOT NODE CLASSIFICATION BY NODE-LEVEL GRAPH DIFFUSION

**Yancheng Wang, Chengshuai Zhao, Dongfang Sun, Huan Liu & Yingzhen Yang**
School of Computing and Augmented Intelligence
Arizona State University
Tempe, AZ 85281, USA
{yancheng.wang,czhao93,huanliu,yingzhen.yang}@asu.edu

## ABSTRACT

In this paper, we propose a novel node-level graph diffusion method with low-rank feature learning for few-shot node classification (FSNC), termed Low-Rank Few-Shot Graph Diffusion Model or LR-FGDM. LR-FGDM first employs a novel Few-Shot Graph Diffusion Model (FGDM) as a node-level graph generative method to generate an augmented graph with an enlarged support set, then performs low-rank transductive classification to obtain the few-shot node classification results. Our graph diffusion model, FGDM, comprises two components, the Hierarchical Graph Autoencoder (HGAE) with an efficient hierarchical edge reconstruction method and a new prototypical regularization, and the Latent Diffusion Model (LDM). The low-rank regularization is robust to the noise inherently introduced by the diffusion model and empirically inspired by the Low Frequency Property. We also provide a strong theoretical guarantee justifying the low-rank regularization for the transductive classification in few-shot learning. To further enhance the performance of LR-FGDM, we introduce LRA-LR-FGDM with a novel efficient LR-Attention layer, or the LRA layer, which applies self-attention to the output of the LR-FGDM encoder. The LRA layer further reduces the kernel complexity of LR-FGDM and contributes to a tighter generalization bound, leading to improved performance. Extensive experimental results evidence the effectiveness of LR-FGDM for few-shot node classification, which outperforms the current state-of-the-art. The code of the LR-FGDM is available at https://github.com/Statistical-Deep-Learning/LR-FGDM.

## 1 INTRODUCTION

Graph Neural Networks (GNNs) (Kipf & Welling, 2016b; Hamilton et al., 2017) are widely used for semi-supervised node classification (Veličković et al., 2018), but their effectiveness relies on ample labeled data. This challenge motivates few-shot node classification (FSNC), where only a few labeled nodes per class are available. Most FSNC methods (Zhou et al., 2019; Ding et al., 2020; Wang et al., 2022; Huang & Zitnik, 2020; Qian et al., 2021; Lan et al., 2020; Liu et al., 2021b) follow a meta-learning framework (Finn et al., 2017; Snell et al., 2017) to generalize across tasks. More recent approaches (Tan et al., 2022; Liu et al., 2024) leverage self-supervised Graph Contrastive Learning (GCL) (Mo et al., 2022; Jin et al., 2021; Wang & Yang, 2022; Ding et al., 2023), and achieve superior performance despite using only unlabeled data. However, all existing methods remain constrained by the limited support set size. Although techniques like mix-up (Liu et al., 2025b) and random perturbation (Wu et al., 2022) offer marginal gains, the potential of generative models to synthesize support nodes remains underexplored. Building on the success of diffusion models in vision, recent works have extended them to synthetic graph generation. Recent works (Niu et al., 2020; Jo et al., 2022; Haefeli et al., 2022; Vignac et al., 2023; Limnios et al., 2023) adapt diffusion models to generate realistic structures that align well with real-world networks. However, these approaches focus on graph-level generation and do not support structured node- or edge-level synthesis. Node-level graph augmentation typically relies on GANs (Jia et al., 2023; Wu et al., 2023; Wang et al., 2018; Liang et al., 2020; Yang et al., 2019) to generate minority class nodes in imbalanced graphs,

despite known issues of training instability and poor distributional matching (Dhariwal & Nichol, 2021).

In this work, we propose a novel node-level graph diffusion method with low-rank feature learning for FSNC, termed Low-Rank Few-Shot Graph Diffusion Model or LR-FGDM. LR-FGDM employs a novel Few-Shot Graph Diffusion Model (FGDM) to generate an augmented graph with an enlarged support set. The FGDM in LR-FGDM consists of two components, including the Hierarchical Graph Autoencoder (HGAE) with an efficient hierarchical edge reconstruction method and the Latent Diffusion Model (LDM). The HGAE learns compact latent node features for LDM by incorporating a prototypical regularization to encourage semantic structure in the latent space. The hierarchical edge reconstruction method enables efficient reconstruction of the edges connecting to a node from the latent space in a hierarchical manner to avoid the quadratic complexity in edge reconstruction of the regular GAE (Kipf & Welling, 2016a). Given a FSNC task, the FGDM generates the synthetic graph structure, consisting of the synthetic support nodes and the edges connecting to the original graph. The synthetic graph structure is then incorporated into the original graph, forming an augmented graph with an enlarged support set consisting of the original and synthetic support nodes.

Although prior methods enlarge the support set via random perturbations (Gao et al., 2023b) or mix-up (Liu et al., 2025b), they fail to generate faithful graph structures, often assigning edges to synthetic nodes by reusing neighbors of real nodes. In contrast, our FGDM jointly encodes nodes and edges into a semantically regularized latent space for training the LDM, capturing the true joint distribution of features and structure. Let $\mathcal{V}_{syn}$ and $\mathcal{V}_{sup}$ denote the set of synthetic support nodes and the original support set. As shown in Figure 1, while adding synthetic support nodes with FGDM improves COLA's performance when $|\mathcal{V}_{syn}| \leq 3|\mathcal{V}_{sup}|$, further increasing the number of synthetic nodes leads to a sharp performance drop. This is due to inherent noise in the diffusion generation process (Ho et al., 2020; Fu et al., 2024; Azizi et al., 2023; He et al., 2022). To this end, we propose a low-rank learning method inspired by the widely-studied Low Frequency Property (LFP) (Rahaman et al., 2019; Arora et al., 2019; Cao et al., 2021; Choraria et al., 2022; Wang et al., 2024; 2025b), which suggests that the pro-

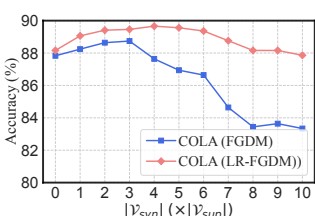

Figure 1: 5-way 5-shot node classification accuracies on Cora-Full trained with different numbers of synthetic support nodes added. COLA (LR-FGDM) is trained on the augmented graph by FGDM with the low-rank regularization, while COLA (FGDM) is trained without the regularization.

jection of the ground truth class labels mostly concentrates on the top eigenvectors of the kernel gram matrix of the model, to be detailed in Section 3.4. Motivated by LFP, the truncated nuclear norm (TNN) is added as a low-rank regularization term to the training loss of the few-shot classifier on the augmented graph. The effectiveness of low-rank learning is also theoretically justified by Theorem A.1 in Section A, which shows that reducing kernel complexity yields a tighter upper bound on the test loss. Inspired by this result, we further propose LRA-LR-FGDM with a low-rank attention (LRA) layer that applies self-attention to the LR-FGDM encoder output, further reducing kernel complexity and tightening the generalization bound compared to LR-FGDM. It is observed from Figure 1 that the COLA trained with the low-rank regularization performs significantly better than the regular COLA when the synthetic nodes added in the augmented graph are more than $3|\mathcal{V}_{sup}|$.

Existing graph few-shot learning methods (Liu et al., 2024; Wang et al., 2023a; Ma et al., 2025; Zhao et al., 2025) show that training graph encoders without label supervision yields better generalization to novel classes. However, the diffusion-based generator DoG (Wang et al., 2025b) relies on class labels as conditioning signals during both training and generation, which is problematic as test-time labels are disjoint from the training labels. Although semi-supervised $K$-means (Basu et al., 2002; Bair, 2013) can be applied on the node attributes to obtain pseudo labels for conditioning, as illustrated in Figure 3 (a), it often leads to semantic drift and unreliable conditioning due to entangled base and novel class semantics. In contrast, our LR-FGDM conditions the diffusion model on cluster prototypes jointly learned with the prototypical regularization, as illustrated in Figure 3 (b), rather than pseudo labels, to avoid this issue. As another significant difference from DoG (Wang et al., 2025b), a new prototypical regularization is introduced to HGAE to improve cluster separability in the latent space, making the prototype-based conditioning semantically aligned and robust.

**Contributions.** The contributions of this paper are presented as follows.

First, we propose the Low-Rank Few-Shot Graph Diffusion Model (LR-FGDM), a novel generative framework for FSNC tasks by synthesizing labeled support nodes and the associated edges through a node-level graph diffusion model, Few-Shot Graph Diffusion Model (FGDM). The FGDM features a new Hierarchical Graph Autoencoder (HGAE) that incorporates prototypical regularization to structure the latent space semantically, and our LDM uses the prototypes instead of class labels such as those in DoG (Wang et al., 2025b) as the conditioning features. To mitigate the inherent inefficiency of the quadratic complexity in edge reconstruction over the entire graph, FGDM also introduces a hierarchical edge reconstruction method, which hierarchically reconstructs the edges connecting to each synthetic support node. While prior methods (Liu et al., 2025b; Wu et al., 2022) have shown promising results in enlarging the support set for FSNC, Table 8 in Section G.2 shows that LR-FGDM substantially outperforms existing support set augmentation approaches. Moreover, we introduce the Frechet Node Distance (FND) and the Frechet Edge Distance (FED) to validate the faithfulness of the synthetic support nodes and the associated edges in Section G.7 of the appendix.

Second, we introduce a low-rank regularization method in LR-FGDM for the transductive node classifier trained on the augmented graph, which is empirically motivated by the Low Frequency Property (LFP) in deep learning (Rahaman et al., 2019; Arora et al., 2019; Cao et al., 2021; Choraria et al., 2022; Wang et al., 2024; 2025b) and theoretically justified by a novel generalization bound for the transductive few-shot node classifier in Theorem A.1. The low-rank regularization promotes lower kernel complexity (KC), thus leading to a lower generalization bound for the test loss of the transductive classifier. Inspired by Theorem A.1, we further introduce LRA-LR-FGDM with a novel efficient LR-Attention layer, or the LRA layer, which applies self-attention to the output of the LR-FGDM encoder. LRA further reduces the KC of LR-FGDM, leading to a tighter generalization bound for the test loss than LR-FGDM. Table 9 in Section G.3 demonstrates that LR-FGDM achieves substantially lower kernel complexity (KC) and a tighter upper bound for the test loss compared to the baseline without low-rank regularization, and LRA-LR-FGDM further reduces KC, leading to an even tighter upper bound for the test loss than LR-FGDM. Furthermore, as shown in Table 1, Table 6, and Table 10, LR-FGDM significantly outperforms state-of-the-art FSNC methods across multiple graph benchmarks, and LRA-LR-FGDM further improves the performance of LR-FGDM.

## 2 RELATED WORKS

### 2.1 FEW-SHOT NODE CLASSIFICATION (FSNC)

While GNNs for node classification are commonly trained in a semi-supervised fashion (Kipf & Welling, 2016b), many efforts (Sun et al., 2020; Hamilton et al., 2017; Veličković et al., 2018) aim to reduce label reliance; however, they struggle with unseen classes at inference, motivating the study of FSNC. Most previous FSNC methods (Zhou et al., 2019; Finn et al., 2017; Yao et al., 2020; Snell et al., 2017; Liu et al., 2024; Wang et al., 2022; Wu et al., 2024; Zhang et al., 2025a) adopt a meta-learning framework by training the FSNC model through a series of meta tasks. More recently, several works have incorporated contrastive learning into meta-learning to enhance task-specific representation learning. (Liu et al., 2021a) and CPLAE (Gao et al., 2021) apply supervised contrastive losses within meta-tasks using augmented views, while PsCo (Jang et al., 2023) and MetaContrastive (Ni et al., 2021) perform unsupervised contrastive meta-learning. COLA (Liu et al., 2024) contrasts support and query prototypes to promote class-level consistency, and COSMIC (Wang et al., 2023a) leverages multi-view contrastive regularization.

### 2.2 GRAPH DIFFUSION MODELS AND GENERATIVE DATA AUGMENTATION ON GRAPH

Score-based diffusion models (Song et al., 2021b) have achieved state-of-the-art performance in diverse generative tasks (Ho et al., 2020; Song & Ermon, 2019; Gao et al., 2023a; Rombach et al., 2022; Baranchuk et al., 2022; Song et al., 2021c;a; Song & Ermon, 2020; Wang et al., 2025a). Graph diffusion models have emerged for synthetic graph generation (Niu et al., 2020; Haefeli et al., 2022; Jo et al., 2022; Zhou et al., 2024), with early works (Jo et al., 2022; Haefeli et al., 2022; Vignac et al., 2023) designing discrete diffusion processes over adjacency matrices. SaGess (Limnios et al., 2023) performs conditional generation of graphs inspired by LDM. However, these models primarily target graph-level generation, limiting their utility in node-level tasks such as FSNC. To enhance the performance of GNNs, node-level data augmentation has been applied to structure (Zhao et al.,

2021b; Rong et al., 2020; Feng et al., 2022; Lai et al., 2024), features (You et al., 2020; Kong et al., 2022; Azad & Fang, 2024), and labels (Han et al., 2022; Wang et al., 2021; Verma et al., 2021; Zhao et al., 2024c). In FSNC, recent methods enhance the support set by perturbing node features and leveraging pseudo-labeled queries (Wu et al., 2022), or by using LLM-based prompting to generate synthetic support nodes for text-attributed graphs (Zhang et al., 2025b). Generative data augmentation has been used to enhance GNN performance by generating synthetic nodes and edges to address class imbalance and enrich minority class features and connectivity (Zhao et al., 2021b; Zhou et al., 2024; Qu et al., 2021; Zhao et al., 2021a; Hsu et al., 2024; Gao et al., 2023b; Hsu et al., 2023). However, these approaches often rely on GANs (Jia et al., 2023; Wu et al., 2023; Wang et al., 2018; Liang et al., 2020; Yang et al., 2019), which suffer from training instability and poor alignment with real data distributions (Dhariwal & Nichol, 2021). To the best of our knowledge, FGDM is among the first to synthesize synthetic graph structures via diffusion models for FSNC.

## 3   FORMULATION

We aim to boost the performance of existing FSNC methods by augmenting the support set in a few-shot task, thereby alleviating the data scarcity in each novel class.

**The Pipeline of Integrating LR-FGDM with an Existing Few-Shot Learning Method.** The proposed LR-FGDM serves as a plug-in module to enhance existing FSNC methods, such as COSMIC (Wang et al., 2023a) and COLA (Liu et al., 2024), by augmenting the support set. LR-FGDM consists of three steps for FSNC: (1) training FGDM, which includes learning a Hierarchical Graph Autoencoder (HGAE) on the original graph and a Latent Diffusion Model (LDM) on its latent space; (2) generating an augmented graph by injecting synthetic support nodes and their edges into the original graph; and (3) applying an existing FSNC method to learn node embeddings, followed by training a low-rank transductive classifier on the augmented support set. Figure 2 (a) illustrates the entire training pipeline with LR-FGDM. Figure 2 (b) illustrates the training of the LR-FGDM, and Figure 4 (b) in the appendix illustrates the generation of the synthetic support nodes by LR-FGDM. LR-FGDM improves few-shot classification by expanding the support set, leveraging the benefits of stronger supervision as in prior augmentation studies (Wu et al., 2022; Liu et al., 2025b).

Our LR-FGDM contains two components, which are the generation of an augmented graph with synthetic support data by FGDM, and few-shot learning with low-rank transductive classification on the augmented graph. Our FGDM features a novel Hierarchical Graph Autoencoder (HGAE) with an efficient hierarchical edge reconstruction method and a new prototypical regularization, detailed in Section 3.2. Then, the generation of an augmented graph with the synthetic support nodes and edges is explained in Section 3.3. The low-rank transductive linear classifier for few-shot classification with theoretical guarantee is detailed in Section 3.4.

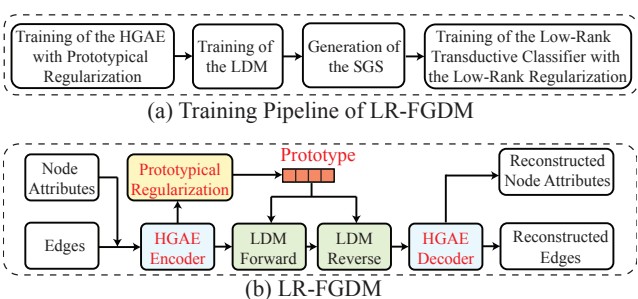

(a) Training Pipeline of LR-FGDM

(b) LR-FGDM

Figure 2: Figure (a) illustrates the entire training pipeline of the FSNC method with LR-FGDM, and Figure (b) illustrates the training of LR-FGDM for synthetic graph structure generation.

### 3.1   PRELIMINARIES

**Few-Shot Node Classification (FSNC).** FSNC assumes disjoint label sets across splits, denoted as $\mathcal{C}_{\text{train}}$, $\mathcal{C}_{\text{val}}$, and $\mathcal{C}_{\text{test}}$ (Liu et al., 2024; Luo et al., 2024; Wang et al., 2023a; Zhao et al., 2024a). An $n$-way task requires the model to classify nodes into $n$ distinct classes randomly sampled from $\mathcal{C}_{\text{test}}$, with only $k$ labeled instances per class provided in the support set. Each task consists of a labeled support set of $n \times k$ nodes and an unlabeled query set. The support set guides the model to learn a transductive classifier to predict the labels of nodes in the query set.

**Attributed Graph and Notations.** An attributed graph with $N'$ nodes is denoted by $\mathcal{G} = (\mathcal{V}, \mathcal{E}, \mathbf{X})$. Here, $\mathcal{V} = \{v_1, v_2, \ldots, v'_N\}$ represents the nodes, and $\mathcal{E} \subseteq \mathcal{V} \times \mathcal{V}$ represents the edges. Node

attributes are given by $\mathbf{X} \in \mathbb{R}^{N' \times D}$, where each row $\mathbf{X}_i \in \mathbb{R}^D$ corresponds to the attributes of node $v_i$. The adjacency matrix $\mathbf{A} \in \{0, 1\}^{N' \times N'}$ of graph $\mathcal{G}$ defines connections. Each row $\mathbf{A}_i$ represents the connections of node $v_i$. The neighborhood $\mathcal{N}(i) = \{j \mid \mathbf{A}_{i,j} = 1\}$ includes node $v_i$ itself and all nodes connected to $v_i$. The notation $[N']$ denotes all natural numbers from 1 to $N'$ inclusive. $[\mathbf{A}]_i$ stands for the $i$-th row of a matrix $\mathbf{A}$. $\|\cdot\|_p$ denotes the $p$-norm of a vector or a matrix.

## 3.2 FEW-SHOT GRAPH DIFFUSION MODEL (FGDM)

**Hierarchical Graph Autoencoder (HGAE) with Prototypical Regularization.** To encode a node $v_i$, we first generate a latent feature of the node attribute $\mathbf{X}_i$ as $f(\mathbf{X}_i)$, where $f(\cdot)$ is a Multi-Layer Perceptron (MLP) layer. To incorporate the information from the edges connected to $v_i$, we add positional embeddings (Ma et al., 2021; You et al., 2019) to the node attributes of $v_i$'s neighbors. For each neighbor $j \in \mathcal{N}(i)$, we modify the node attributes as $\mathbf{X}'_j = \mathbf{X}_j + \text{pos}(j)$, where $\text{pos}(\cdot)$ is a function converting the position index into an embedding vector (Vaswani et al., 2017). We apply two Graph Attention Network (GAT) (Veličković et al., 2018) layers to aggregate the information in $\{\mathbf{X}'_j \mid j \in \mathcal{N}(i)\}$ into a single latent feature $\mathbf{Z}'_i$. Next, we concatenate $\mathbf{Z}'_i$ with $f(\mathbf{X}_i)$ to obtain the latent feature of node $v_i$ by $\mathbf{Z}_i = f'(\mathbf{Z}'_i \| f(\mathbf{X}_i))$, where $f'$ is another MLP layer encoding the concatenated features to the latent space of LDM with lower dimension $D'$. After encoding a node $v_i$ in the graph to a latent feature $\mathbf{Z}_i$, the decoder of the HGAE reconstructs the node attribute $\widehat{\mathbf{X}}_i$ by three consecutive MLP layers and its associated edges $\widehat{\mathbf{A}}_i$ by the hierarchical edge reconstruction method to be introduced later.

*Prototypical Regularization.* Existing FSNC methods (Snell et al., 2017; Laenen & Bertinetto, 2021; Ding et al., 2020; Lin et al., 2022) show that prototypical learning improves node embeddings by promoting intra-class compactness and inter-class separability. To align latent features of semantically similar nodes, we add a prototypical regularization to the HGAE loss, encouraging nodes within the same cluster to approach shared prototypes. The cluster assignments are obtained via semi-supervised $K$-means (Basu et al., 2002; Bair, 2013) utilizing labeled nodes to guide clustering while incorporating unlabeled nodes for better generalization to unseen classes. Let $\mathbf{p}_c \in \mathbb{R}^{D'}$ represent the prototype of cluster $c \in [K]$, where $K$ is the number of prototypes. The prototypical regularization loss is defined as $\mathcal{L}_{\text{proto}} = \sum_{i=1}^{N'} \left\| \mathbf{Z}_i - \mathbf{p}_{\pi(i)} \right\|^2$, where $\mathbf{p}_{\pi(i)} = \frac{1}{|\mathcal{V}_{\pi(i)}|} \sum_{j \in \mathcal{V}_{\pi(i)}} \mathbf{Z}_j$. $\pi(i)$ is the cluster index of node $v_i$ and $\mathcal{V}_{\pi(i)}$ is the set of nodes in the cluster $\pi(i)$.

To address the quadratic complexity inherent in conventional GAEs (Zhai et al., 2018; Kipf & Welling, 2016a), we introduce a prototype-guided hierarchical edge reconstruction framework designed to promote efficient edge decoding. The edge reconstruction is conducted hierarchically based on clusters induced by learned prototype representations. We define an inter-cluster neighbor map $\mathbf{C} \in \{0, 1\}^{N' \times K}$, where $\mathbf{C}_{ik} = 1$ indicates that node $v_i$ connects to at least one node within the cluster represented by prototype $k$. Additionally, an intra-cluster neighbor map $\mathbf{M} \in \{0, 1\}^{N' \times K \times S}$ is constructed, where $\mathbf{M}_{iks} = 1$ signifies that node $v_i$ is connected to the $s$-th node within cluster $k$, with $S$ denoting the maximum number of nodes in any cluster. In contrast to the Bi-Level Neighborhood Decoder (BLND) employed in DoG (Wang et al., 2025b) using balanced $K$-means applied to node attributes, our method leverages prototype clusters learned jointly with the encoder, because nodes in the same prototype cluster have similar latent features, thus tend to connect with each other. The structure of the network used for the hierarchical edge reconstruction in the HGAE with prototypical regularization is illustrated in Figure 5 in Section F of the appendix.

**Training the HGAE with Prototypical Regularization.** For each node $v_i$, the hierarchical edge reconstruction method first reconstructs its inter-cluster neighbor map $\widehat{\mathbf{C}}_i$ with one MLP layer. After that, the predicted cluster indices $\mathcal{C}(i) = \{k \in [K] | \mathbf{C}_{ik} = 1\}$ are separately fed to an embedding layer to generate a set of class-conditional features $\mathcal{Z}(i) = \left\{ g(k) \in \mathbb{R}^{D'} | k \in \mathcal{C}(i) \right\}$ using the class-conditional embedding method in Classifier-Free Guidance (Ho & Salimans, 2022), where $g$ contains one text embedding layer followed by an MLP layer. Next, each of the class-conditional features $g(k) \in \mathcal{Z}(i)$ is concatenated with the latent feature of the other branch for decoding the intra-cluster neighbor map by $\widehat{\mathbf{M}}_{ik} = g'(\mathbf{Z}_i \| g(k))$, where $g'$ is another MLP layer. The HGAE is trained by minimizing the sum of the node reconstruction loss, the hierarchical edge reconstruction

loss, and the prototypical loss $\mathcal{L}_{\text{proto}}$ as follows,

$$\mathcal{L}_{\text{HGAE}} = \underbrace{\left\|\mathbf{X} - \widehat{\mathbf{X}}\right\|_2^2}_{\text{Node Reconstruction Loss}} + \underbrace{\left(\left\|\mathbf{C} - \widehat{\mathbf{C}}\right\|_2^2 + \left\|\mathbf{M} - \widehat{\mathbf{M}}\right\|_2^2\right)}_{\text{Hierarchical Edge Reconstruction Loss}} + \mathcal{L}_{\text{proto}}, \qquad (1)$$

where $\|\cdot\|_2$ denotes the Euclidean norm. We perform a detailed complexity analysis of the hierarchical edge reconstruction method in Section C of the appendix. Table 3 in Section 4.4 of the appendix demonstrates the improved efficiency of the proposed hierarchical edge reconstruction method compared to the decoder in a regular GAE.

**Training the LDM.** Once the HGAE with the hierarchical edge reconstruction method is trained, we obtain a set of latent representations $\mathbf{Z} = \{\mathbf{Z}_i \in \mathbb{R}^{D'} \mid v_i \in \mathcal{V}\}$ encoding both node attributes and edges. Traditional class-conditional diffusion models typically condition on class labels, including DoG (Wang et al., 2025b), as illustrated in Figure 3 (a). However, in FSNC, the classes in the support and query sets are novel and disjoint from those used during training. As the diffusion model is trained prior to test-time adaptation, it cannot directly condition on these unseen class labels. To overcome this limitation, we leverage the prototypical regularization introduced in the HGAE, which encourages the latent representations $\mathbf{Z}_i$ to cluster around their respective prototype representations. These prototypes, computed as the mean latent representation of each cluster, serve as semantically meaningful and continuous conditioning signals. Instead of relying on discrete class labels, the LDM is conditioned directly on the corresponding prototype representation for each training node, enabling prototype-based conditional generation. As illustrated in Figure 3 (b), each latent feature is paired with its assigned prototype as the conditioning input under the Classifier-Free Guidance (CFG) framework (Ho & Salimans, 2022). This design allows the LDM to learn to generate latent features aligned with the semantic structure of the data without requiring access to class labels. The training algorithm of FGDM is presented in Algorithm 1 in Section H of the appendix.

### 3.3 GENERATION OF AUGMENTED GRAPH WITH SYNTHETIC SUPPORT DATA BY FGDM

**Generation of Synthetic Graph Structures with FGDM.** Once the FGDM is trained, we aim to generate synthetic graph structures consisting of synthetic support nodes and edges connecting to the original graph. We first obtain the cluster label of each of the support nodes obtained from the prototypical regularization, which is used to get the prototype representation for the conditional generation of the synthetic graph structure, as illustrated in Figure 4 (b) in the appendix. Let $\mathcal{V}_{\text{sup}} \subseteq \mathcal{V}$ be the original support nodes and $\mathcal{V}_{\text{syn}}$ be the synthetic support nodes. Let the node attributes of $\mathcal{V}_{\text{syn}}$ be $\mathbf{X}_{\text{syn}}$ and the affinity matrix encoding edges between the synthetic nodes and real nodes be $\mathbf{A}_{\text{syn}}$. Then the synthetic graph structure is denoted as $(\mathcal{V}_{\text{syn}}, \mathbf{X}_{\text{syn}}, \mathbf{A}_{\text{syn}})$. Let $M$ be the number of nodes in the synthetic graph structure. The adjacency matrix of the augmented graph is $\mathbf{A}_{\text{aug}} = \left[\mathbf{A} \ \mathbf{A}_{\text{syn}}; \mathbf{A}_{\text{syn}}^\top \ \mathbf{0}\right] \in \mathbb{R}^{(N'+M) \times (N'+M)}$, and the node attributes of the augmented graph is $\mathbf{X}_{\text{aug}} = [\mathbf{X}; \mathbf{X}_{\text{syn}}] \in \mathbb{R}^{(N'+M) \times D}$. The augmented graph, which is the combination of the original graph $\mathcal{G}$ and the synthetic graph structure, is then denoted by $\mathcal{G}_{\text{aug}} = (\mathcal{V} \cup \mathcal{V}_{\text{syn}}, \mathbf{X}_{\text{aug}}, \mathbf{A}_{\text{aug}})$. Let $\mathcal{V}_{\mathcal{L}} = \mathcal{V}_{\text{sup}} \cup \mathcal{V}_{\text{syn}}$ denote the augmented support set. For a $n$-way $k$-shot FSNC task, we generate the synthetic graph structures, consisting of $M = q \times n \times k$ synthetic support nodes and their edges connecting to the original graph, where $q$ denotes the number of synthetic nodes generated per real support node. The value of $q$ for different tasks on different datasets is selected by cross-validation as detailed in Section G.5. The augmented support set $\mathcal{V}_{\mathcal{L}}$ then consists of $(q+1)nk$ support nodes with $(q+1)k$ nodes in each of the $n$ novel classes. The augmented graph $\mathcal{G}_{\text{aug}}$ is then encoded using existing few-shot graph encoders, such as COSMIC (Wang et al., 2023a) and COLA (Liu et al., 2024), yielding the representation for all the nodes in the augmented graph, which is denoted as $\mathbf{H} \in \mathbb{R}^{(N'+M) \times d}$. The generation of the augmented graph is described in Algorithm 2 in Section H of the appendix.

### 3.4 LOW-RANK TRANSDUCTIVE LINEAR CLASSIFIER FOR FEW-SHOT LEARNING

Due to the inherent stochasticity of diffusion models (Ho et al., 2020; Rombach et al., 2022), the synthetic graph structures generated by LR-FGDM may introduce noise, leading to semantic mismatches between synthetic support nodes and their labels (Azizi et al., 2023; He et al., 2022). To address this, we follow prior FSNC methods (Wang et al., 2023a; Liu et al., 2024) by training a transductive node classifier on embeddings from a few-shot graph encoder. Motivated by the Low

Frequency Property (LFP) (Rahaman et al., 2019; Arora et al., 2019; Cao et al., 2021; Choraria et al., 2022; Wang et al., 2024; 2025b), which suggests that class labels concentrate on top eigenvectors of the model's kernel gram matrix, we introduce a novel low-rank regularization for the classifier with theoretical guarantees.

**Notation Definition.** Let $\mathbf{u} \in \mathbb{R}^{N'}$ be a vector, we use $[\mathbf{u}]_{\mathcal{A}}$ to denote a vector formed by elements of $\mathbf{u}$ with indices in $\mathcal{A}$ for $\mathcal{A} \subseteq [N']$. If $\mathbf{u}$ is a matrix, then $[\mathbf{u}]_{\mathcal{A}}$ denotes a submatrix formed by rows of $\mathbf{u}$ with row indices in $\mathcal{A}$. $\|\cdot\|_{\mathrm{F}}$ denotes the Frobenius norm, and $\|\cdot\|_p$ denotes the $p$-norm. Let $\mathcal{V}_{\mathrm{FS}} = \mathcal{V}_{\mathcal{L}} \cup \mathcal{V}_{\mathcal{U}}$ denote all the nodes from the $n$ novel classes in an $n$-way $k$-shot task, where $\mathcal{V}_{\mathcal{L}}$ and $\mathcal{V}_{\mathcal{U}}$ are the labeled support set and the unlabeled query set, respectively, with $|\mathcal{V}_{\mathcal{L}}| = m$. Let $N$ denote the number of nodes in $\mathcal{V}_{\mathrm{FS}}$. $\mathcal{L}$ and $\mathcal{U}$ denote the indices of the nodes in $\mathcal{V}_{\mathcal{L}}$ and $\mathcal{V}_{\mathcal{U}}$. Let $\mathcal{V}_{\mathrm{FS}} = \{v'_1, v'_2, \dots, v'_N\}$, where $v'_i$ is the $i$-th node in $\mathcal{V}_{\mathrm{FS}}$. Let $\mathbf{y}_i \in \mathbb{R}^n$ be the ground-truth one-hot class label vector for $v'_i$ in $\mathcal{V}_{\mathrm{FS}}$, and define $\mathbf{Y}_{\mathrm{FS}} \coloneqq [\mathbf{y}_1; \mathbf{y}_2; \dots \mathbf{y}_N] \in \mathbb{R}^{N \times n}$ be the ground-truth label matrix defined on the $n$ novel classes for all the nodes in $\mathcal{V}_{\mathrm{FS}}$. Let $\mathbf{H}_{\mathrm{FS}} \in \mathbb{R}^{N \times d}$ be the representations of all the nodes in $\mathcal{V}_{\mathrm{FS}}$. We define $\mathbf{F}(\mathbf{W}) = \mathbf{H}_{\mathrm{FS}} \mathbf{W}$ as the linear output of the transductive few-shot classifier with $\mathbf{W} \in \mathbb{R}^{d \times n}$ being the weight matrix. Let $\mathbf{K}$ be the gram matrix of the node representations $\mathbf{H}_{\mathrm{FS}}$, which is calculated by $\mathbf{K} = \mathbf{H}_{\mathrm{FS}} \mathbf{H}_{\mathrm{FS}}^{\top}/N \in \mathbb{R}^{N \times N}$. Let $\left\{\widehat{\lambda}_i\right\}_{i=1}^N$ with $\widehat{\lambda}_1 \geq \widehat{\lambda}_2 \dots \geq \widehat{\lambda}_{\min\{N,d\}} \geq \widehat{\lambda}_{\min\{N,d\}+1} = \dots = 0$ be the eigenvalues of $\mathbf{K}$.

**Low-Rank Transductive Few-Shot Node Classification.** In order to encourage the features $\mathbf{H}_{\mathrm{FS}}$ or the gram matrix $\mathbf{K}$ to be low-rank, we explicitly add the truncated nuclear norm $\|\mathbf{K}\|_{r_0} \coloneqq \sum_{i=r_0+1}^N \widehat{\lambda}_i$ to the loss function of the transductive few-shot node classifier. The starting rank $r_0 < \min(N, d)$ is the rank of the features $\mathbf{H}_{\mathrm{FS}}$ we aim to keep in the node representation, that is, if $\|\mathbf{K}\|_{r_0} = 0$, then $\mathrm{rank}(\mathbf{K}) = r_0$. Therefore, the overall loss function is

$$\min_{\mathbf{W}} L(\mathbf{W}) = \frac{1}{m} \sum_{i:\, v'_i \in \mathcal{V}_{\mathcal{L}}} \mathrm{KL}\left(\mathbf{y}_i, [\mathrm{softmax}\left(\mathbf{H}_{\mathrm{FS}} \mathbf{W}\right)]_i\right) + \tau \|\mathbf{K}\|_{r_0}, \tag{2}$$

where KL is the KL divergence. $\tau > 0$ is the weighting parameter for the truncated nuclear norm $\|\mathbf{K}\|_{r_0}$. We use a regular gradient descent to optimize (2) with a learning rate $\eta \in (0, \frac{1}{\widehat{\lambda}_1})$. $\mathbf{W}$ is initialized by $\mathbf{W}^{(0)} = \mathbf{0}$, and at the $t$-th iteration of gradient descent for $t \geq 1$, $\mathbf{W}$ is updated by $\mathbf{W}^{(t)} = \mathbf{W}^{(t-1)} - \eta \nabla_{\mathbf{W}} L(\mathbf{W})|_{\mathbf{W}=\mathbf{W}^{(t-1)}}$. The optimal rank $r_0$ on different datasets is decided by cross-validation as detailed in Section 4.1 of the appendix. Here $\|\mathbf{K}\|_{r_0}$ can be efficiently computed by the Nystrom method detailed in Section I of the appendix.

**Motivation of the Low-Rank Regularization.** We study how the information of the ground-truth class label defined on the novel classes is distributed on different eigenvectors of the feature gram matrix $\mathbf{K}$ by performing eigen-projection in Section G.9 of the appendix. It is observed in Figure 6 in Section G.9 of the appendix that the projection of the ground truth labels for the novel classes mostly concentrates on the top eigenvectors of $\mathbf{K}$, known as the Low Frequency Property (LFP) widely studied in other areas of machine learning (Rahaman et al., 2019; Arora et al., 2019; Cao et al., 2021; Choraria et al., 2022; Wang et al., 2024; 2025b). *We remark that the low-rank regularization ensures that mostly only the low-rank part of the node representations $\mathbf{H}_{\mathrm{FS}}$ is used for the FSNC, so that our transductive node classifier trained by (2) is free of the noise in the high-rank part of the $\mathbf{H}_{\mathrm{FS}}$, thus being robust to the noise in the synthetic graph structures introduced by LR-FGDM.* The low-rank learning is also theoretically justified by Theorem A.1 in Section A of the appendix, showing that the low-rank learning reduces the kernel complexity and renders a tighter bound for the test loss.

### 3.5 LRA-LR-FGDM: Improving LR-FGDM by Low Rank Attention

To further improve the performance of LR-FGDM, we introduce LRA-LR-FGDM with a novel LR-Attention layer, or the LRA layer, which applies self-attention to the output of the LR-FGDM encoder by $\mathbf{F} = \mathbf{B} \mathbf{H}_{\mathrm{FS}}$, where $\mathbf{H}_{\mathrm{FS}} \in \mathbb{R}^{N \times d}$ is the low-rank node representations produced by the LR-FGDM encoder. $\mathbf{F}$ is the attention output and $\mathbf{B} \in \mathbb{R}^{N \times N}$ is our new attention matrix in the LRA layer. We recall that the kernel gram matrix of the node features $\mathbf{H}_{\mathrm{FS}}$ is $\mathbf{K} = \mathbf{H}_{\mathrm{FS}} \mathbf{H}_{\mathrm{FS}}^{\top}/N$. The attention weight matrix $\mathbf{B}$ is set to $\mathbf{B} = \mathbf{K}/\widehat{\lambda}_1$.

**Efficiency of LRA with the Low-Rank Attention Weight Matrix B.** The attention matrix in LRA is computed by $\mathbf{B} = \mathbf{K}/\widehat{\lambda}_1$, which implies $\text{rank}(\mathbf{B}) \leq \min\{N, d\}$. In practice, the embedding dimension $d$ is significantly smaller than the number of nodes $N$. In all our experiments, we set $d = 256$, following standard settings in existing FSNC methods (Wang et al., 2023b; Liu et al., 2024). As a result, $\mathbf{B}$ is inherently low-rank, and the LRA operation $\mathbf{B}\mathbf{H}_{\text{FS}} = \mathbf{H}_{\text{FS}}\mathbf{H}_{\text{FS}}^{\top}\mathbf{H}_{\text{FS}}/\widehat{\lambda}_1$ can be computed in $\mathcal{O}(Nd^2)$ time by first computing the $d \times d$ matrix $\mathbf{H}_{\text{FS}}^{\top}\mathbf{H}_{\text{FS}}$ and then multiplying it with $\mathbf{H}_{\text{FS}}$, without explicitly constructing the dense $N \times N$ attention matrix.

The gram matrix $\mathbf{K}_{\mathbf{F}}$ of the node representations $\mathbf{F} \in \mathbb{R}^{N \times d}$ is then $\mathbf{K}_{\mathbf{F}} = \mathbf{F}\mathbf{F}^{\top} = \mathbf{K}^3/\widehat{\lambda}_1^2$. Let $\{\lambda_i\}_{i=1}^{N}$ be the eigenvalues of $\mathbf{K}_{\mathbf{F}}$ with $\lambda_1 \geq \lambda_2 \geq ...\lambda_N \geq 0$, then we have $\lambda_i = \widehat{\lambda}_i^3/\widehat{\lambda}_1^2$ for every $i \in [N]$. Noting that $\lambda_i = \widehat{\lambda}_i \cdot \widehat{\lambda}_i^2/\widehat{\lambda}_1^2 \leq \widehat{\lambda}_i$ due to the fact that $\widehat{\lambda}_1 \geq \widehat{\lambda}_i$ for all $i \in [N]$, therefore, the LRA layer reduces the kernel complexity of the kernel gram matrix $\mathbf{K}$, because the KC of $\mathbf{K}_{\mathbf{F}}$ is always not greater than that of $\mathbf{K}$. We then train a transductive classifier on top of $\mathbf{F}$ similar to Section 3.4 by minimizing the loss function

$$\min_{\mathbf{W}} L(\mathbf{W}) = \frac{1}{m} \sum_{i:\, v_i' \in \mathcal{V}_{\mathcal{L}}} \text{KL}\left(\mathbf{y}_i, [\text{softmax}(\mathbf{FW})]_i\right) + \tau\|\mathbf{K}_{\mathbf{F}}\|_{r_0}. \quad (3)$$

Such a linear classifier trained with the LRA layer through the optimization of (3) is termed LRA-LR-FGDM. It then follows from the above discussion and the upper bound for the test loss (5) in Theorem A.1 that LRA-LR-FGDM has a lower KC, so that the test loss $\mathcal{U}_{\text{test}}(t)$ of LRA-LR-FGDM can be even lower than that of LR-FGDM, suggesting a better prediction accuracy of LRA-LR-FGDM than LR-FGDM. This is empirically justified in Table 9 where LRA-LR-FGDM exhibits lower KC and lower upper bound for the test loss than that of LR-FGDM.

## 4 EXPERIMENTS

We evaluate the performance of the LR-FGDM for shot augmentation combined with the low-rank regularization for FSNC. In Section 4.1, we present the implementation details of the proposed LR-FGDM. In Section 4.2, we present the results for different FSNC settings. An ablation study on the effectiveness of the prototypical regularization in the HGAE and low-rank regularization on the few-shot classifier is performed in Section 4.3. In Section 4.4, we perform the efficiency analysis of LR-FGDM. Additional experiment results are presented in Section G of the appendix. In Section G.1, we present the results for FSNC on three additional graph datasets. We also compare the LR-FGDM against existing state-of-the-art shot augmentation methods in Section G.2. In Section G.3, we study the effectiveness of LR-FGDM in reducing the kernel complexity of the kernel gram matrix and the upper bound for the test loss of the transductive linear classifier in LR-FGDM. In Section G.4, we study the effectiveness of LR-FGDM on a heterophilic graph dataset, the Roman-Empire dataset (Platonov et al., 2023). In Section G.5, we present the details about cross-validation used to select the number of synthetic support nodes. In Section G.6, we perform the sensitivity analysis of the hyperparameters $\tau$, $r_0$, $K$, and $q$. We have proposed the Frechet Node Distance (FND) and the Frechet Edge Distance (FED) to validate the faithfulness of the synthetic support nodes and the associated edges with comparison to existing shot augmentation methods in Section G.7. The statistical significance of the improvements achieved by LR-FGDM in Section 4.2 and Section 4.3 is validated by the student $t$-test detailed in Section G.8. In Section G.9, we analyze how the ground-truth label signal distributes across the eigenvectors of the feature Gram matrix. In Section G.10, we study an automatic eigengap-based strategy for selecting $r_0$. In Section G.11, we provide a t-SNE visualization comparing the embeddings of synthetic nodes generated by LR-FGDM with those of real nodes in the novel classes. In Section G.12, we conduct an ablation study to evaluate the impact of the LDM. In Section G.13, we report the training time of LR-FGDM and LRA-LR-FGDM on the augmented graphs. In Section G.14, we evaluate whether augmented graphs generated by LR-FGDM preserve key graph structural properties. In our experiments, we apply LR-FGDM on top of existing FSNC methods, COSMIC (Wang et al., 2023a) and COLA (Liu et al., 2024), which are the most recent state-of-the-art FSNC methods with the best performance.

### 4.1 IMPLEMENTATION DETAILS

We conduct experiments for FSNC on CoraFull (Bojchevski & Günnemann, 2018), ogbn-arxiv (Hu et al., 2020), Coauthor-CS (Shchur et al., 2018), DBLP (Tang et al., 2008), Roman-Empire (Platonov

et al., 2023), Amazon-Computers, Amazon-Photo (Shchur et al., 2018), and Citeseer (Sen et al., 2008), with details in Section D.1 of the appendix. The training settings of LR-FGDM and the hyper-parameter tuning are described in Section D.2 of the appendix.

Table 1: The overall FSNC results of all methods under different settings. The best result is in bold, and the second-best result is underlined. The statistical significance of the results is deferred to Table 17 of the appendix.

| Dataset | CoraFull | | | | ogbn-arxiv | | | |
|---|---|---|---|---|---|---|---|---|
| Task | 2-way 1-shot | 2-way 5-shot | 5-way 1-shot | 5-way 5-shot | 2-way 1-shot | 2-way 5-shot | 5-way 1-shot | 5-way 5-shot |
| ProtoNet (Snell et al., 2017) | $57.10 \pm 2.47$ | $72.71 \pm 2.55$ | $32.43 \pm 1.61$ | $51.54 \pm 1.68$ | $62.56 \pm 2.86$ | $75.82 \pm 2.79$ | $37.30 \pm 2.00$ | $53.31 \pm 1.71$ |
| Meta-GNN (Zhou et al., 2019) | $75.28 \pm 3.85$ | $84.59 \pm 2.89$ | $55.33 \pm 2.43$ | $70.50 \pm 2.02$ | $62.52 \pm 3.41$ | $70.15 \pm 2.68$ | $27.14 \pm 1.94$ | $31.52 \pm 1.71$ |
| GPN (Ding et al., 2020) | $74.29 \pm 3.47$ | $85.58 \pm 2.53$ | $52.75 \pm 2.32$ | $72.82 \pm 1.88$ | $64.00 \pm 3.71$ | $76.78 \pm 3.50$ | $37.81 \pm 2.34$ | $50.50 \pm 2.13$ |
| G-Meta (Huang & Zitnik, 2020) | $78.23 \pm 3.41$ | $89.49 \pm 2.04$ | $60.44 \pm 2.48$ | $75.84 \pm 1.70$ | $63.03 \pm 3.32$ | $76.56 \pm 2.89$ | $31.48 \pm 1.70$ | $47.16 \pm 1.73$ |
| TENT (Wang et al., 2022) | $77.75 \pm 3.29$ | $88.20 \pm 2.61$ | $55.44 \pm 2.08$ | $70.10 \pm 1.73$ | $70.30 \pm 2.85$ | $81.35 \pm 2.77$ | $48.26 \pm 1.73$ | $61.38 \pm 1.72$ |
| KD-FSNC (Wu et al., 2024) | $83.92 \pm 2.68$ | $94.08 \pm 2.42$ | $74.55 \pm 2.47$ | $85.89 \pm 2.15$ | $74.86 \pm 3.15$ | $84.67 \pm 2.39$ | $52.74 \pm 2.13$ | $64.91 \pm 1.70$ |
| NormProp (Zhang et al., 2025a) | $83.61 \pm 2.64$ | $93.87 \pm 2.39$ | $74.21 \pm 2.52$ | $85.47 \pm 2.14$ | $74.33 \pm 3.10$ | $84.36 \pm 2.41$ | $52.37 \pm 2.11$ | $64.28 \pm 1.72$ |
| STAR (Liu et al., 2025a) | $85.22 \pm 1.69$ | $94.95 \pm 1.48$ | $75.85 \pm 1.72$ | $87.31 \pm 1.55$ | $76.45 \pm 2.03$ | $86.11 \pm 2.10$ | $54.82 \pm 1.75$ | $66.98 \pm 1.25$ |
| DoG (Wang et al., 2025b) | $85.10 \pm 1.98$ | $94.35 \pm 1.82$ | $75.13 \pm 1.56$ | $86.47 \pm 1.13$ | $77.33 \pm 2.31$ | $86.89 \pm 2.21$ | $53.42 \pm 1.47$ | $65.69 \pm 1.85$ |
| COSMIC (Wang et al., 2023a) | $84.32 \pm 2.75$ | $94.51 \pm 2.47$ | $74.93 \pm 2.49$ | $86.34 \pm 2.17$ | $75.71 \pm 3.17$ | $85.19 \pm 2.35$ | $53.28 \pm 2.19$ | $65.42 \pm 1.69$ |
| COLA (Liu et al., 2024) | $85.83 \pm 1.92$ | $95.17 \pm 1.85$ | $76.47 \pm 2.12$ | $87.83 \pm 1.89$ | $77.12 \pm 2.36$ | $86.42 \pm 2.28$ | $55.24 \pm 2.04$ | $67.52 \pm 1.75$ |
| **COSMIC (LR-FGDM)** | $86.21 \pm 2.38$ | $96.74 \pm 2.11$ | $76.93 \pm 2.15$ | $88.81 \pm 1.93$ | $77.68 \pm 2.75$ | $87.24 \pm 2.13$ | $55.48 \pm 2.01$ | $67.59 \pm 1.52$ |
| **COSMIC (LRA-LR-FGDM)** | $86.78 \pm 2.07$ | $97.29 \pm 1.88$ | $77.44 \pm 1.95$ | $89.47 \pm 1.72$ | $78.33 \pm 2.14$ | $87.89 \pm 1.96$ | $56.02 \pm 1.73$ | $68.11 \pm 1.49$ |
| **COLA (LR-FGDM)** | $87.54 \pm 1.74$ | $97.38 \pm 1.67$ | $78.52 \pm 1.94$ | $89.66 \pm 1.72$ | $79.02 \pm 2.18$ | $88.34 \pm 2.10$ | $57.28 \pm 1.86$ | $69.63 \pm 1.57$ |
| **COLA (LRA-LR-FGDM)** | $\mathbf{88.14 \pm 1.62}$ | $\mathbf{97.97 \pm 1.55}$ | $\mathbf{79.17 \pm 1.80}$ | $\mathbf{90.32 \pm 1.59}$ | $\mathbf{79.62 \pm 1.93}$ | $\mathbf{89.03 \pm 1.84}$ | $\mathbf{57.88 \pm 1.71}$ | $\mathbf{70.22 \pm 1.48}$ |

| Dataset | Coauthor-CS | | | | DBLP | | | |
|---|---|---|---|---|---|---|---|---|
| Task | 2-way 1-shot | 2-way 5-shot | 5-way 1-shot | 5-way 5-shot | 2-way 1-shot | 2-way 5-shot | 5-way 1-shot | 5-way 5-shot |
| ProtoNet (Snell et al., 2017) | $59.92 \pm 2.70$ | $71.69 \pm 2.51$ | $32.13 \pm 1.52$ | $49.25 \pm 1.50$ | $60.97 \pm 2.56$ | $72.81 \pm 2.73$ | $31.31 \pm 1.58$ | $52.26 \pm 1.88$ |
| Meta-GNN (Zhou et al., 2019) | $85.90 \pm 2.96$ | $90.11 \pm 2.17$ | $52.86 \pm 2.14$ | $68.59 \pm 1.49$ | $82.60 \pm 3.23$ | $86.15 \pm 3.29$ | $67.24 \pm 2.72$ | $72.15 \pm 2.40$ |
| GPN (Ding et al., 2020) | $84.31 \pm 2.73$ | $90.36 \pm 1.90$ | $60.66 \pm 2.07$ | $81.79 \pm 1.18$ | $79.55 \pm 3.46$ | $85.85 \pm 2.61$ | $59.38 \pm 2.40$ | $75.46 \pm 1.87$ |
| G-Meta (Huang & Zitnik, 2020) | $84.19 \pm 2.97$ | $91.02 \pm 1.61$ | $59.68 \pm 2.16$ | $74.18 \pm 1.29$ | $80.46 \pm 3.29$ | $88.53 \pm 2.36$ | $63.32 \pm 2.70$ | $75.82 \pm 2.11$ |
| TENT (Wang et al., 2022) | $87.85 \pm 2.48$ | $91.75 \pm 1.60$ | $63.70 \pm 1.88$ | $76.90 \pm 1.19$ | $84.40 \pm 2.73$ | $90.05 \pm 2.34$ | $61.56 \pm 2.23$ | $74.84 \pm 2.04$ |
| KD-FSNC (Wu et al., 2024) | $89.78 \pm 2.36$ | $93.21 \pm 2.01$ | $67.05 \pm 1.66$ | $84.42 \pm 1.17$ | $91.81 \pm 2.41$ | $94.37 \pm 1.70$ | $74.83 \pm 2.15$ | $83.75 \pm 1.91$ |
| NormProp (Zhang et al., 2025a) | $89.34 \pm 2.41$ | $93.62 \pm 1.97$ | $67.48 \pm 1.68$ | $84.61 \pm 1.14$ | $91.52 \pm 2.45$ | $94.05 \pm 1.72$ | $75.39 \pm 2.18$ | $84.12 \pm 1.89$ |
| STAR (Liu et al., 2025a) | $91.28 \pm 1.15$ | $95.41 \pm 1.85$ | $69.25 \pm 1.23$ | $87.60 \pm 1.33$ | $93.10 \pm 1.47$ | $95.52 \pm 1.55$ | $77.14 \pm 1.35$ | $87.10 \pm 1.05$ |
| DoG (Wang et al., 2025b) | $91.10 \pm 1.84$ | $94.88 \pm 1.53$ | $68.96 \pm 1.80$ | $87.35 \pm 1.41$ | $93.55 \pm 1.35$ | $96.05 \pm 1.22$ | $78.87 \pm 1.37$ | $87.59 \pm 1.25$ |
| COSMIC (Wang et al., 2023a) | $90.29 \pm 2.30$ | $94.32 \pm 1.93$ | $68.21 \pm 1.63$ | $85.47 \pm 1.11$ | $92.35 \pm 2.52$ | $94.82 \pm 1.69$ | $76.52 \pm 2.24$ | $85.31 \pm 1.92$ |
| COLA (Liu et al., 2024) | $91.53 \pm 2.03$ | $95.78 \pm 1.84$ | $70.46 \pm 1.57$ | $87.54 \pm 1.19$ | $93.48 \pm 2.17$ | $95.92 \pm 1.68$ | $78.18 \pm 2.05$ | $87.23 \pm 1.87$ |
| **COSMIC (LR-FGDM)** | $92.48 \pm 2.01$ | $96.71 \pm 1.67$ | $70.41 \pm 1.48$ | $87.72 \pm 1.03$ | $94.78 \pm 2.29$ | $96.95 \pm 1.53$ | $78.66 \pm 2.03$ | $87.44 \pm 1.71$ |
| **COSMIC (LRA-LR-FGDM)** | $92.94 \pm 1.88$ | $97.24 \pm 1.59$ | $70.90 \pm 1.36$ | $88.21 \pm 1.12$ | $95.33 \pm 2.10$ | $97.44 \pm 1.41$ | $79.11 \pm 1.92$ | $87.96 \pm 1.58$ |
| **COLA (LR-FGDM)** | $93.84 \pm 1.85$ | $97.91 \pm 1.56$ | $72.93 \pm 1.41$ | $89.83 \pm 1.11$ | $95.89 \pm 2.03$ | $97.98 \pm 1.47$ | $80.16 \pm 1.88$ | $89.51 \pm 1.65$ |
| **COLA (LRA-LR-FGDM)** | $\mathbf{94.33 \pm 1.71}$ | $\mathbf{98.41 \pm 1.49}$ | $\mathbf{73.46 \pm 1.29}$ | $\mathbf{90.39 \pm 1.03}$ | $\mathbf{96.41 \pm 1.92}$ | $\mathbf{98.52 \pm 1.39}$ | $\mathbf{80.65 \pm 1.75}$ | $\mathbf{90.07 \pm 1.57}$ |

## 4.2 RESULTS

We compare the performance of the proposed LR-FGDM with state-of-the-art FSNC methods, including ProtoNet (Snell et al., 2017), Meta-GNN (Zhou et al., 2019), GPN (Ding et al., 2020), G-Meta (Huang & Zitnik, 2020), TENT (Wang et al., 2022), KD-FSNC (Wu et al., 2024), Norm-Prop (Zhang et al., 2025a), COSMIC (Wang et al., 2023a), COLA (Liu et al., 2024), and STAR (Liu et al., 2025a). We also compare LR-FGDM with the diffusion-based synthetic graph structure generation method DoG (Wang et al., 2025b). Since DoG requires label-conditioning during training and generation, which is unavailable for unseen classes in few-shot settings, we employ semi-supervised $K$-means (Basu et al., 2002; Bair, 2013) to obtain pseudo labels as conditioning signals for DoG. The number of clusters and the number of synthetic nodes are all decided by cross-validation. We integrate LR-FGDM and LRA-LR-FGDM into COSMIC and COLA, resulting in four variants, denoted as COSMIC (LR-FGDM), COLA (LR-FGDM), COSMIC (LRA-LR-FGDM), and COLA (LRA-LR-FGDM). The experiments are conducted for 2-way and 5-way classification tasks, each with 1-shot and 5-shot settings following (Liu et al., 2024; Wang et al., 2023a). The mean accuracy and standard deviation across 20 independent runs for each setting are reported. It is observed in Table 1 that LR-FGDM consistently improves the performance of COSMIC and COLA on all the datasets. For example, COLA (LR-FGDM) outperforms COLA by 2.29% on Coauthor-CS for the 5-way 5-shot FSNC. LRA-LR-FGDM models show further improved performance over our LR-FGDM models, which significantly outperform the COSMIC and COLA. The results on the heterophilic graph dataset, Roman-Empire, and three additional graph datasets, Amazon-Computers, Amazon-Photo, and Citeseer, are deferred to Table 10 and Table 6 in the appendix.

## 4.3 ABLATION STUDY

To thoroughly study the effectiveness of the prototypical regularization in the HGAE and low-rank regularization on the classifier, we conduct an ablation study on CoraFull, ogbn-arxiv, Coauthor-CS, and DBLP under the 5-way 5-shot setting for FSNC. We evaluate three variants of the COLA (LR-FGDM), which are the COLA (LR-FGDM) without the prototypical regularization, COLA (LR-FGDM) without the low-rank regularization, and COLA (LR-FGDM) without both the low-rank regularization and the prototypical regularization. It is observed from Table 2 that both the

low-rank regularization on training the few-shot node classifier and the prototypical regularization on training the HGAE play important roles in improving the performance of the baseline method.

Table 2: Ablation study on the low-rank regularization and the prototypical regularization. The study is performed under the 5-way 5-shot setting for FSNC. The statistical significance of the results is deferred to Table 18 of the appendix.

| Method | CoraFull | ogbn-arxiv | Coauthor-CS | DBLP |
|---|---|---|---|---|
| COLA (Liu et al., 2024) | 87.83 | 67.52 | 87.54 | 87.23 |
| COLA (LR-FGDM) w/o both low-rank and prototypical regularization | 88.12 | 67.91 | 87.93 | 87.55 |
| COLA (LR-FGDM) w/o low-rank regularization | 88.74 | 68.60 | 88.72 | 88.28 |
| COLA (LR-FGDM) w/o prototypical regularization | 88.79 | 68.45 | 89.02 | 88.64 |
| COLA (LR-FGDM) | 89.66 | 69.63 | 89.83 | 89.51 |
| COLA (LRA-LR-FGDM) | **90.32** | **70.22** | **90.39** | **90.07** |

## 4.4 TRAINING EFFICIENCY ANALYSIS

To study the efficiency of the FGDM, we compare the training time between our HGAE with hierarchical edge reconstruction and the regular GAE without hierarchical edge reconstruction. In addition, we also compare the time for the generation of our FGDM and FGDM without hierarchical edge reconstruction in its HGAE. All evaluations are conducted using a single Nvidia A100 GPU. It is observed from the results in Table 3 that the hierarchical edge reconstruction method significantly reduces the computation cost of the training and synthetic graph structure generation. For instance, the training of GAE without hierarchical edge reconstruction takes over five times the training time of our GAE with hierarchical edge reconstruction on ogbn-arxiv. In addition, the hierarchical edge reconstruction method also significantly reduces the time for synthetic graph structure generation. For instance, the data generation without the hierarchical edge reconstruction method takes over four times the data generation time of our FGDM on ogbn-arxiv.

Table 3: Time for the training of GAE and LDM in FGDM and data generation with FGDM on different datasets. The training time for the FSNC models on the augmented graphs generated by FGDM is shown in Table 22 of the appendix.

| Datasets | Training Time (seconds) | | | Generation Time (s/sample) | |
|---|---|---|---|---|---|
| | HGAE | GAE w/o Hierarchical Edge Reconstruction | LDM | FGDM | FGDM w/o Hierarchical Edge Reconstruction |
| CoraFull | 41 | 129 | 154 | 0.067 | 0.073 |
| Coauthor-CS | 52 | 145 | 179 | 0.074 | 0.088 |
| ogbn-arxiv | 301 | 1690 | 315 | 0.130 | 0.426 |
| DBLP | 11 | 16 | 39 | 0.049 | 0.066 |

## 5 CONCLUSION

In this paper, we propose a novel node-level graph diffusion method with low-rank feature learning for FSNC, termed Low-Rank Few-Shot Graph Diffusion Model or LR-FGDM. LR-FGDM addresses the limitation of data scarcity in few-shot settings by augmenting the support set through a novel node-level graph diffusion model and enforcing low-rank regularization on the training of the few-shot node classifier. FGDM integrates a Hierarchical Graph Autoencoder (HGAE) with a hierarchical edge reconstruction method and a Latent Diffusion Model (LDM). The low-rank regularization is motivated by the Low Frequency Property (LFP) and theoretically justified by a theorem to show lower generalization error. Inspired by the novel theorem, we further introduce LRA-LR-FGDM with a novel LR-Attention layer, or the LRA layer, which further reduces the KC of LR-FGDM, leading to a tighter generalization bound for the test loss than LR-FGDM. Extensive experiments on multiple graph benchmark datasets show that LR-FGDM significantly improves the performance of few-shot node classifiers, demonstrating superior generalization capabilities compared to state-of-the-art methods.

## ACKNOWLEDGMENTS

Y. Wang and Y. Yang are supported by the 2023 Mayo Clinic and Arizona State University Alliance for Health Care Collaborative Research Seed Grant Program under Award No. AWD00038846 and by the NIH grant under Award No. 1OT2OD037955-01. C. Zhao and H. Liu are supported by the U.S. National Science Foundation (NSF) under grant IIS-2229461.

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

## A    THEORETICAL JUSTIFICATION FOR THE LOW-RANK REGULARIZATION

We have the following theoretical result, Theorem A.1, on the Mean Squared Error (MSE) loss of the unlabeled query nodes $\mathcal{U}$ measured by the gap between $[\mathbf{F}(\mathbf{W}, t)]_{\mathcal{U}}$ and $[\mathbf{Y}_{\text{FS}}]_{\mathcal{U}}$ when using the low-rank feature $\mathbf{H}_{\text{FS}}$ with $r_0 \in [N]$, which is the generalization error bound for the linear transductive classifier using $\mathbf{F}(\mathbf{W}) = \mathbf{H}_{\text{FS}}\mathbf{W}$ to predict the labels of the query nodes. Similar to existing works such as (Kothapalli et al., 2023) that use the MSE to analyze the optimization and the generalization of GNNs, we employ the MSE loss to provide the generalization error of the node classifier in the following theorem. It is remarked that the MSE loss is necessary for the generalization analysis of transductive learning using transductive local Rademacher complexity (Tolstikhin et al., 2014; Yang, 2025a;b). We use $\mathbf{A}^{(i)}$ to denote the $i$-th column of a matrix $\mathbf{A}$, and suppose that the eigendecomposition of $\mathbf{K}$ is $\mathbf{K} = \mathbf{U}\boldsymbol{\Sigma}\mathbf{U}^{\top}$, where $\mathbf{U}$ is an orthogonal matrix and $\boldsymbol{\Sigma}$ is a diagonal matrix with its diagonal elements being $\left\{\widehat{\lambda}_i\right\}_{i=1}^{N}$. $\widehat{\lambda}_i$ is an eigenvalue of $\mathbf{K}$ with $\mathbf{U}^{(i)}$ as the corresponding eigenvector for all $i \in [N]$.

**Theorem A.1.** Suppose that $\mathcal{V}_{\mathcal{L}}$ is sampled uniformly without replacement from $\mathcal{V}_{\text{FS}}$, and $\mathcal{V}_{\mathcal{U}} = \mathcal{V}_{\text{FS}} \setminus \mathcal{V}_{\mathcal{L}}$ is the query set containing the test nodes where $\mathcal{U}$ is the set of the indices of nodes in $\mathcal{V}_{\mathcal{U}}$, where $|\mathcal{V}_{\mathcal{L}}| = m$ and $|\mathcal{V}_{\mathcal{U}}| = u$. Suppose that for each $j \in [n]$, the $j$-th column of $\mathbf{Y}_{\text{FS}}$, $\mathbf{Y}_{\text{FS}}^{(j)}$, satisfies

$$\sum_{i=1}^{N} \frac{\left[\mathbf{U}^{\top}\mathbf{Y}_{\text{FS}}^{(j)}\right]_i^2}{\widehat{\lambda}_i} \leq \mu^2 \tag{4}$$

for some positive number $\mu$. Then after the $t$-th iteration of gradient descent on the training loss $\tilde{L}(\mathbf{W}) \coloneqq 1/(2m) \cdot \sum_{v_i' \in \mathcal{V}_{\mathcal{L}}} \|\mathbf{y}_i - [\mathbf{H}_{\text{FS}}\mathbf{W}]_i\|_2^2$ for all $t \geq 1$, for every $x > 0$ and every $\delta \in (0, 1)$, with probability at least $1 - \exp(-x) - \delta$, we have

$$\mathcal{U}_{\text{test}}(t) \coloneqq \frac{1}{u}\|[\mathbf{F}(\mathbf{W}, t) - \mathbf{Y}_{\text{FS}}]_{\mathcal{U}}\|_{\text{F}}^2 \leq \frac{L(\mathbf{K}, \mathbf{Y}_{\text{FS}}, t)}{m} + c_0 \text{KC}(\mathbf{K}) + \frac{c_0 x}{N_{u,m,\delta}}, \tag{5}$$

where

$$N_{u,m,\delta} \coloneqq \frac{\min\{u, m\}}{\log_2(4\min\{u, m\}/\delta)},$$

$c_0$ is a positive number depending on $n$, $\mu$, and $\tau_0$ with $\tau_0^2 = \max_{i \in [N]} \mathbf{K}_{ii}$. Moreover, $L(\mathbf{K}, \mathbf{Y}_{\text{FS}}, t) := \left\| (\mathbf{I}_m - \eta \mathbf{K}_{\mathcal{L},\mathcal{L}})^t [\mathbf{Y}_{\text{FS}}]_{\mathcal{L}} \right\|_{\text{F}}^2$, $\mathbf{K}_{\mathcal{L},\mathcal{L}} := [\mathbf{H}_{\text{FS}}]_{\mathcal{L}} [\mathbf{H}_{\text{FS}}]_{\mathcal{L}}^\top / m$, and KC is the kernel complexity of the kernel gram matrix $\mathbf{K} = \mathbf{H}_{\text{FS}} \mathbf{H}_{\text{FS}}^\top / N$ defined as $\text{KC}(\mathbf{K}) := \min_{r_0 \in [N]} r_0 \left( \frac{1}{u} + \frac{1}{m} \right) + \sqrt{\|\mathbf{K}\|_{r_0}} \left( \frac{1}{\sqrt{u}} + \frac{1}{\sqrt{m}} \right)$.

This theorem is proved in Section B of the appendix. Detailed explanation about Theorem A.1 is deferred to Section B.1 of the appendix.

## B  THEORETICAL RESULTS

We present the proof of Theorem A.1 in this section.

**Proof of Theorem A.1.** It can be verified that at the $t$-th iteration of gradient descent for $t \geq 1$, we have

$$\mathbf{W}^{(t)} = \mathbf{W}^{(t-1)} - \frac{\eta}{m} [\mathbf{H}_{\text{FS}}]_{\mathcal{L}}^\top \left[ \mathbf{H}_{\text{FS}} \mathbf{W}^{(t-1)} - \mathbf{Y}_{\text{FS}} \right]_{\mathcal{L}}. \tag{6}$$

It follows from (6) that

$$[\mathbf{H}_{\text{FS}}]_{\mathcal{L}} \mathbf{W}^{(t)} = [\mathbf{H}_{\text{FS}}]_{\mathcal{L}} \mathbf{W}^{(t-1)} - \eta \mathbf{K}_{\mathcal{L},\mathcal{L}} \left[ \mathbf{H}_{\text{FS}} \mathbf{W}^{(t-1)} - \mathbf{Y}_{\text{FS}} \right]_{\mathcal{L}}, \tag{7}$$

where $\mathbf{K}_{\mathcal{L},\mathcal{L}} = [\mathbf{H}_{\text{FS}}]_{\mathcal{L}} [\mathbf{H}_{\text{FS}}]_{\mathcal{L}}^\top / m \in \mathbb{R}^{m \times m}$. With $\mathbf{F}(\mathbf{W}, t) = \mathbf{H}_{\text{FS}} \mathbf{W}^{(t)}$, it follows from (7) that

$$[\mathbf{F}(\mathbf{W}, t) - \mathbf{Y}_{\text{FS}}]_{\mathcal{L}} = (\mathbf{I}_m - \eta \mathbf{K}_{\mathcal{L},\mathcal{L}}) [\mathbf{F}(\mathbf{W}, t-1) - \mathbf{Y}_{\text{FS}}]_{\mathcal{L}}.$$

It then follows from the above recursion that

$$[\mathbf{F}(\mathbf{W}, t) - \mathbf{Y}_{\text{FS}}]_{\mathcal{L}} = - (\mathbf{I}_m - \eta \mathbf{K}_{\mathcal{L},\mathcal{L}})^t [\mathbf{Y}_{\text{FS}}]_{\mathcal{L}}. \tag{8}$$

It can be verified that

$$[\mathbf{F}(\mathbf{W}, t)]_{ij} = \Theta(\tau_0 \mu), \quad \forall i \in [N], j \in [n].$$

We apply (Yang, 2025b, Eq. (B.70)) to obtain the following bound for the test loss $\frac{1}{u} \| [\mathbf{F}(\mathbf{W}, t) - \mathbf{Y}_{\text{FS}}]_{\mathcal{U}} \|_{\text{F}}^2$:

$$\frac{1}{u} \| [\mathbf{F}(\mathbf{W}, t) - \mathbf{Y}_{\text{FS}}]_{\mathcal{U}} \|_{\text{F}}^2 \leq \frac{1}{m} \| [\mathbf{F}(\mathbf{W}, t) - \mathbf{Y}_{\text{FS}}]_{\mathcal{L}} \|_{\text{F}}^2 + c_0 \min_{0 \leq Q \leq n} r(u, m, Q) + \frac{c_0 x}{N_{u,m,\delta}}, \tag{9}$$

with

$$r(u, m, Q) := Q \left( \frac{1}{u} + \frac{1}{m} \right) + \left( \sqrt{\frac{\sum_{q=Q+1}^{N} \widehat{\lambda}_q}{u}} + \sqrt{\frac{\sum_{q=Q+1}^{N} \widehat{\lambda}_q}{m}} \right),$$

where $c_0$ is a positive number depending on $n$, $\mu$, and $\tau_0$ with $\tau_0^2 = \max_{i \in [N]} \mathbf{K}_{ii}$.

It follows from (8) and (9) that for every $r_0 \in [N]$, we have

$$\frac{1}{u} \| [\mathbf{F}(\mathbf{W}, t) - \mathbf{Y}_{\text{FS}}]_{\mathcal{U}} \|_{\text{F}}^2 \leq \frac{1}{m} \left\| (\mathbf{I}_m - \eta \mathbf{K}_{\mathcal{L},\mathcal{L}})^t [\mathbf{Y}_{\text{FS}}]_{\mathcal{L}} \right\|_{\text{F}}^2$$

$$+ c_0 r_0 \left( \frac{1}{u} + \frac{1}{m} \right) + c_0 \left( \sqrt{\frac{\sum_{q=r_0+1}^{N} \widehat{\lambda}_q}{u}} + \sqrt{\frac{\sum_{q=r_0+1}^{N} \widehat{\lambda}_q}{m}} \right) + \frac{c_0 x}{N_{u,m,\delta}}$$

$$\overset{\textcircled{1}}{\leq} \frac{1}{m} \left\| (\mathbf{I}_m - \eta \mathbf{K}_{\mathcal{L},\mathcal{L}})^t [\mathbf{Y}_{\text{FS}}]_{\mathcal{L}} \right\|_{\text{F}}^2 + c_0 r_0 \left( \frac{1}{u} + \frac{1}{m} \right) + c_0 \sqrt{\|\mathbf{K}\|_{r_0}} \left( \sqrt{\frac{1}{u}} + \sqrt{\frac{1}{m}} \right) + \frac{c_0 x}{N_{u,m,\delta}}, \tag{10}$$

where ① (8) and $\sum_{q=r_0+1}^{N} \widehat{\lambda}_q = \|\mathbf{K}\|_{r_0}$. (5) then follows directly from (10). $\qquad\square$

### B.1 FURTHER EXPLANATION OF THEOREM A.1

Define $\mathbf{F}(\mathbf{W}, t) := \mathbf{H}_{\text{FS}}\mathbf{W}^{(t)}$ as the output of the classifier after the $t$-th iteration of gradient descent for $t \geq 1$. It is noted that $\mathcal{U}_{\text{test}}(t)$ is the test loss of the unlabeled query nodes measured by the distance between the classifier output $\mathbf{F}(\mathbf{W}, t)$ and $\mathbf{Y}_{\text{FS}}$. There are two terms on the upper bound for the test loss in (5), $L(\mathbf{K}, \mathbf{Y}_{\text{FS}}, t)$ and $KC(\mathbf{K})$, which are explained as follows. $L(\mathbf{K}, \mathbf{Y}_{\text{FS}}, t)$ corresponds to the training loss of the node classifier with the ground-truth label for the novel classes. $KC(\mathbf{K})$ is the kernel complexity (KC), which measures the complexity of the kernel gram matrix from the node representation $\mathbf{H}_{\text{FS}}$. We remark that the TNN $\|\mathbf{K}\|_{r_0}$ appears on the RHS of the upper bound (5), theoretically justifying why we learn the low-rank features $\mathbf{K}$ for FSNC by adding the TNN $\|\mathbf{K}\|_{r_0}$ to the training loss. Moreover, when the low frequency property holds, $L(\mathbf{K}, \mathbf{Y}_{\text{FS}}, t)$ would be very small with enough iteration number $t$. $\mathbf{K} = \mathbf{H}_{\text{FS}}^{\top}\mathbf{H}_{\text{FS}}$ is approximately a low-rank matrix of rank $r_0$ since $\mathbf{H}_{\text{FS}}$ is approximately a rank-$r_0$ matrix with its TNN optimized through the optimization of the encoder of the HGAE. A smaller $\|\mathbf{K}\|_{r_0}$ is obtained by optimizing the training loss in Equation (2), which in turn ensures a smaller kernel complexity (KC) defined in Theorem A.1, contributing to a smaller generalization bound for transductive node classification.

## C COMPLEXITY ANALYSIS OF THE HIERARCHICAL EDGE RECONSTRUCTION METHOD

In our work, we have proposed an efficient hierarchical edge reconstruction method to reconstruct the edges connected to a node in the graph. To show its efficiency, we analyze the inference time complexity and the parameter size of the HGAE with the hierarchical edge reconstruction method. For comparison, we also analyze the inference time complexity and the parameter size of GAE, where the hierarchical edge reconstruction method is replaced by a regular edge decoder that directly reconstructs the adjacency matrix $\mathbf{A}$ (Kipf & Welling, 2016a). For ease of comparison, we denote the number of parameters and inference cost of all the MLP and GAT layers except the hierarchical edge reconstruction process as $S_{\text{MLP}}$ and $C_{\text{MLP}}$, respectively. For a node $v_i$ in the graph, let $d_i = \sum_{k=1}^{K} \widehat{\mathbf{C}}_{ik}$ be the number of clusters predicted to be connected to $v_i$. Let $D'$ be the dimension of the input feature for the hierarchical edge reconstruction. The inference time complexity of HGAE with hierarchical edge reconstruction is $\mathcal{O}(KD' + d_iD'M + C_{\text{MLP}})$, where $\mathcal{O}(KD')$ is the additional complexity for computing the inter-cluster neighbor map and encoding the cluster indices. $\mathcal{O}(d_iD'M)$ is the computation cost for computing the intra-cluster neighbor map. In contrast, the inference time complexity of a regular GAE with a regular edge decoder is $\mathcal{O}(D'KM + C_{\text{MLP}})$. We note that $d_i$ is upper bounded by the degree of the node $v_i$. In most graph datasets, the average degree of nodes is usually very small. For instance, on CoraFull, where the average node degree is 6.41, we have $d_i \leq 6.41$. As a result, $D'(K + d_iM) \ll D'KM$. For example, setting $K = 200$ and $M = 100$ on Pubmed, we find that the inference time complexity of HGAE with hierarchical edge reconstruction is $\mathcal{O}(841D' + C_{\text{MLP}})$, which is much more efficient than the regular edge decoder whose inference time complexity is $\mathcal{O}(20000D' + C_{\text{MLP}})$. In general, the inference time complexity of HGAE with hierarchical edge reconstruction is much lower than that of GAE with a regular edge decoder.

Table 4: Statistics of the graph datasets.

| Dataset | # Nodes | # Edges | # Features | # Classes |
|---|---|---|---|---|
| CoraFull | 19,793 | 63,421 | 8,710 | 70 |
| ogbn-arxiv | 169,343 | 1,166,243 | 128 | 40 |
| Coauthor-CS | 18,333 | 81,894 | 6,805 | 15 |
| DBLP | 40,672 | 144,135 | 7,202 | 137 |
| Roman-Empire | 22,662 | 32,927 | 64 | 18 |
| Citeseer | 3,327 | 4,732 | 3,703 | 6 |
| Amazon-Computers | 13,752 | 245,861 | 767 | 10 |
| Amazon-Photo | 7,650 | 119,081 | 745 | 8 |

## D    ADDITIONAL EXPERIMENT DETAILS

### D.1    DATASETS

To evaluate the performance of our method on FSNC, we conduct experiments on eight widely used real-world benchmark datasets, which are CoraFull (Bojchevski & Günnemann, 2018), ogbn-arxiv (Hu et al., 2020), Coauthor-CS (Shchur et al., 2018), DBLP (Tang et al., 2008), Roman-Empire (Platonov et al., 2023), Amazon-Computers, Amazon-Photo (Shchur et al., 2018), and Cite-seer (Sen et al., 2008) with their statistics summarized in Table 4. CoraFull is an extended version of the Cora dataset, constructed from the entire citation network, where nodes represent papers and edges denote citation links; node classes correspond to paper topics. ogbn-arxiv is a directed ci-tation graph derived from the arXiv Computer Science category in the Microsoft Academic Graph (MAG) (Wang et al., 2020a), where nodes are arXiv papers and edges represent citation relations. Node labels are based on 40 CS subject areas. Coauthor-CS is a co-authorship graph extracted from MAG during the KDD Cup 2016 challenge, where nodes denote authors and edges indicate co-authorship. Node features are derived from paper keywords, and node classes correspond to the authors' most active research fields. DBLP is another citation network in which nodes represent papers and edges denote citation links. Node features are based on paper abstracts, and labels corre-spond to publication venues. Roman-Empire is a synthetic dependency graph designed to simulate extreme heterophily where adjacent nodes often belong to different classes. Nodes represent words and edges reflect syntactic dependencies, with class labels assigned based on grammatical roles. Amazon-Computers and Amazon-Photo are two product co-purchase networks from the Amazon dataset, where nodes represent products and edges indicate frequently co-purchased items. Node features are extracted from product reviews, and class labels represent product categories. Citeseer is a citation network where nodes are scientific publications and edges represent citation links. Node features are TF-IDF weighted word vectors, and classes correspond to research topics.

### D.2    TRAINING SETTINGS OF HGAE AND LDM

The training of the HGAE is divided into two phases. In the first phase, we pre-train the HGAE by only minimizing the node reconstruction loss and the edge reconstruction loss in Equation (1) for 500 epochs. In the second phase, we minimize $\mathcal{L}_{\text{HGAE}}$ with the prototypical loss for another 500 epochs. We use the Adam optimizer with a learning rate of 0.001 for the training. The weight decay is set to $1 \times 10^{-5}$. We train the LDM in the LR-FGDM after finishing the training of the HGAE. We use the Adam optimizer with a learning rate of 0.0002 to train the LDM for 1000 epochs.

The rank parameter $r_0$ and the weighting parameter $\tau$ associated with the TNN loss are selected through cross-validation tailored to each dataset. We define the rank as $r_0 = \lceil \gamma \min\{N, d\} \rceil$, where $\gamma$ represents the rank ratio and $d$ is the dimension of the learned node representations. The hyperparameter $\gamma$ is searched over the set $\{0.1, 0.2, 0.3, 0.4, 0.5, 0.6, 0.7, 0.8, 0.9\}$, while the TNN weight $\tau$ is chosen from $\{0.05, 0.1, 0.15, 0.2, 0.25, 0.3, 0.35, 0.4, 0.45, 0.5\}$. The number of the prototype clusters, $K$, is selected from $\{5, 10, 15, 20, 25\}$.

## E    ILLUSTRATION OF THE SYNTHETIC SUPPORT NODE GENERATION BY LR-FGDM

Figure 3 illustrates the encoding and decoding process of the node attributes and the associated edges by the LR-FGDM and the DoG (Wang et al., 2025b), with the difference between LR-FGDM and DoG marked in red.

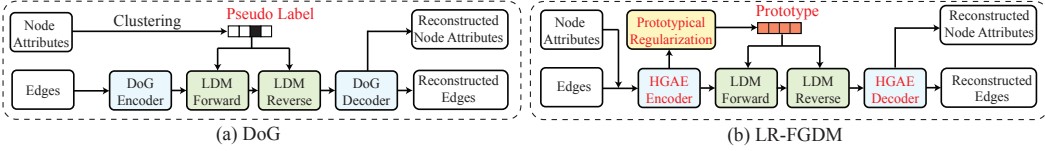

Figure 3: Figure (a) and (b) illustrate the training of the DoG (Wang et al., 2025b) and the training of the LR-FGDM, respectively, with the difference between LR-FGDM and DoG marked in red.

Figure 4 illustrates the generation of the synthetic support nodes and the associated synthetic edges by the LR-FGDM and the DoG (Wang et al., 2025b), with the difference between LR-FGDM and DoG marked in red.

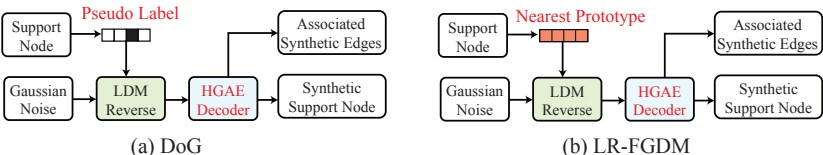

(a) DoG         (b) LR-FGDM

Figure 4: Figure (a) illustrates the generation of the synthetic support nodes and the associated synthetic edges by the DoG (Wang et al., 2025b). Figure (b) illustrates the generation of the synthetic support nodes and the associated synthetic edges by the LR-FGDM.

# F   DETAILS AND STUDIES ON THE HIERARCHICAL EDGE RECONSTRUCTION METHOD

Figure 5 illustrates the structure of the network used for the hierarchical edge reconstruction in the HGAE with prototypical regularization. In contrast to the Bi-Level Neighborhood Decoder (BLND) employed in DoG (Wang et al., 2025b) using balanced $K$-means on node attributes, our method leverages prototype cluster assignment and prototype representations learned jointly with the encoder of the HGAE with prototypical regularization, because nodes in the same prototype cluster have similar latent features, thus tend to connect with each other.

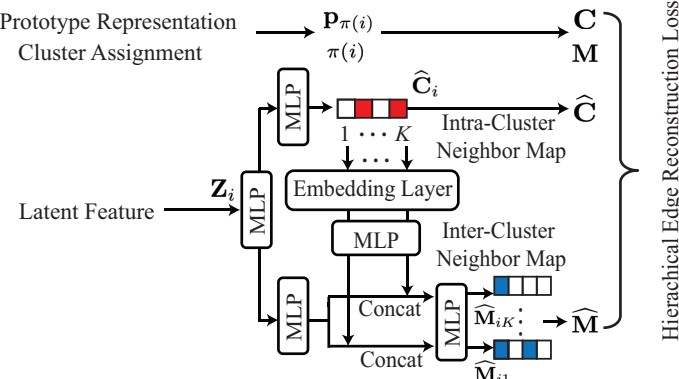

Figure 5: Illustration of the network architecture for the hierarchical edge reconstruction method in the HGAE with prototypical regularization.

To validate the effectiveness of the hierarchical edge reconstruction method compared to the BLND proposed in DoG (Wang et al., 2025b), we perform an ablation study by comparing the performance of the LR-FGDM with an ablation model where the hierarchical edge reconstruction module is replaced by the BLND in DoG (Wang et al., 2025b). The ablation model is denoted as LR-FGDM (BLND). The study is performed for the 5-way 5-shot FSNC task on CoraFull, ogbn-arxiv, Coauthor-CS, and DBLP. It is observed in Table 5 that LR-FGDM consistently outperforms LR-FGDM (BLND) across all datasets. For example, LR-FGDM achieves a $1.16\%$ improvement on ogbn-arxiv, highlighting the superiority of the proposed hierarchical edge reconstruction method in capturing meaningful structural patterns for few-shot learning. These results underscore the benefits of leveraging prototype-guided inter-cluster and intra-cluster connectivity over purely attribute-based neighborhood decoders like BLND.

Table 5: Performance comparison between the proposed hierarchical edge reconstruction method and the Bi-Level Neighborhood Decoder (BLND) in DoG (Wang et al., 2025b) for the 5-way 5-shot FSNC task.

| Data | CoraFull | ogbn-arxiv | Coauthor-CS | DBLP |
|---|---|---|---|---|
| LR-FGDM (BLND) | 88.52 | 68.47 | 88.96 | 88.37 |
| LR-FGDM | **89.66** | **69.63** | **89.83** | **89.51** |

## G  ADDITIONAL EXPERIMENT RESULTS

### G.1  FEW-SHOT NODE CLASSIFICATION ON CITESEER, AMAZON COMPUTERS, AND AMAZON PHOTOS

In this section, we further validate the effectiveness of LR-FGDM for FSNC on additional datasets, including Citeseer (Sen et al., 2008), Amazon Computers (Shchur et al., 2018), and Amazon Photos (Shchur et al., 2018). Due to the limited number of classes in these three datasets, we follow (Wu et al., 2024) and only perform FSNC under the 2-way 1-shot and the 2-way 5-shot settings. It is observed in Table 6 that LR-FGDM consistently improves the performance of COSMIC and COLA on all three datasets and significantly outperforms all competing FSNC methods. To demonstrate the statistical significance of the improvements achieved by LR-FGDM over the baseline methods, we perform $t$-tests on the few-shot classification accuracies obtained from 20 independent few-shot tasks for each setting and each dataset. It is observed in Table 7 that models enhanced by LR-FGDM consistently yield statistically significant improvements over the corresponding methods across all few-shot settings with p-values $p < 0.05$.

Table 6: The overall FSNC results of all methods under different settings for Amazon-Computers, Amazon-Photo, and Citeseer. The node classification accuracy and its standard deviation are in %. The best result under each setting is in bold, and the second-best result is underlined.

| Method | Amazon-Computers | | Amazon-Photo | | Citeseer | |
|---|---|---|---|---|---|---|
| | 2-way 1-shot | 2-way 5-shot | 2-way 1-shot | 2-way 5-shot | 2-way 1-shot | 2-way 5-shot |
| ProtoNet (Snell et al., 2017) | $56.67 \pm 2.54$ | $63.11 \pm 2.60$ | $66.74 \pm 2.08$ | $72.64 \pm 1.94$ | $67.39 \pm 1.65$ | $79.02 \pm 2.33$ |
| Meta-GNN (Zhou et al., 2019) | $60.54 \pm 2.79$ | $68.36 \pm 2.15$ | $69.34 \pm 2.03$ | $76.20 \pm 1.87$ | $67.41 \pm 1.60$ | $79.08 \pm 2.27$ |
| GPN (Ding et al., 2020) | $63.85 \pm 2.31$ | $71.02 \pm 2.07$ | $72.35 \pm 1.92$ | $77.88 \pm 1.74$ | $69.12 \pm 1.68$ | $80.02 \pm 2.14$ |
| G-Meta (Huang & Zitnik, 2020) | $62.56 \pm 3.11$ | $71.47 \pm 2.97$ | $70.18 \pm 2.10$ | $77.45 \pm 1.81$ | $65.53 \pm 1.58$ | $78.01 \pm 1.80$ |
| TENT (Wang et al., 2022) | $77.74 \pm 3.16$ | $86.06 \pm 2.16$ | $84.62 \pm 2.78$ | $86.53 \pm 2.00$ | $75.03 \pm 2.81$ | $85.31 \pm 2.42$ |
| KD-FSNC (Wu et al., 2024) | $86.92 \pm 1.74$ | $95.30 \pm 0.85$ | $91.08 \pm 2.17$ | $96.60 \pm 0.41$ | $79.48 \pm 2.62$ | $86.43 \pm 1.32$ |
| NormProp (Zhang et al., 2025a) | $85.10 \pm 2.08$ | $94.35 \pm 1.30$ | $90.42 \pm 2.21$ | $96.15 \pm 0.53$ | $78.41 \pm 2.37$ | $85.60 \pm 1.61$ |
| COSMIC (Wang et al., 2023a) | $87.12 \pm 1.82$ | $95.60 \pm 1.01$ | $91.54 \pm 2.04$ | $96.12 \pm 0.42$ | $79.77 \pm 2.20$ | $86.23 \pm 1.53$ |
| COLA (Liu et al., 2024) | $87.52 \pm 1.78$ | $95.89 \pm 1.02$ | $91.74 \pm 1.04$ | $96.38 \pm 0.33$ | $80.13 \pm 2.11$ | $87.02 \pm 1.30$ |
| **COSMIC (LR-FGDM)** | $88.63 \pm 1.70$ | $96.74 \pm 0.97$ | $92.38 \pm 1.91$ | $97.22 \pm 0.29$ | $80.92 \pm 2.01$ | $87.62 \pm 1.27$ |
| **COSMIC (LRA-LR-FGDM)** | $89.12 \pm 1.58$ | $\underline{97.25 \pm 0.88}$ | $92.87 \pm 1.79$ | $\underline{97.71 \pm 0.27}$ | $81.38 \pm 1.86$ | $88.15 \pm 1.19$ |
| **COLA (LR-FGDM)** | $\underline{89.14 \pm 1.66}$ | $97.21 \pm 0.89$ | $\underline{93.41 \pm 1.75}$ | $97.45 \pm 0.26$ | $\underline{81.73 \pm 1.84}$ | $\underline{88.21 \pm 1.23}$ |
| **COLA (LRA-LR-FGDM)** | $\mathbf{89.63 \pm 1.55}$ | $\mathbf{97.74 \pm 0.82}$ | $\mathbf{93.92 \pm 1.63}$ | $\mathbf{97.96 \pm 0.25}$ | $\mathbf{82.25 \pm 1.71}$ | $\mathbf{88.74 \pm 1.16}$ |

Table 7: $p$-values from $t$-tests comparing COSMIC (LR-FGDM), COLA (LR-FGDM), COSMIC (LRA-LR-FGDM) and COLA (LRA-LR-FGDM) against their corresponding baseline methods, COSMIC and COLA, in Table 6.

| Dataset | Amazon-Computers | | Amazon-Photo | | Citeseer | |
|---|---|---|---|---|---|---|
| Task | 2-way 1-shot | 2-way 5-shot | 2-way 1-shot | 2-way 5-shot | 2-way 1-shot | 2-way 5-shot |
| COSMIC (LR-FGDM) | 0.026 | 0.018 | 0.021 | 0.012 | 0.034 | 0.017 |
| COLA (LR-FGDM) | 0.008 | 0.004 | 0.006 | 0.003 | 0.010 | 0.005 |
| COSMIC (LRA-LR-FGDM) | 0.004 | 0.003 | 0.007 | 0.002 | 0.004 | 0.010 |
| COLA (LRA-LR-FGDM) | 0.003 | 0.001 | 0.001 | 0.001 | 0.003 | 0.002 |

### G.2  COMPARISON WITH EXISTING SHOT AUGMENTATION METHODS

In this section, we compare LR-FGDM with existing shot augmentation methods. IA-FSNC (Wu et al., 2022) incorporates confidently predicted query nodes as additional labeled instances and introduces noise-based perturbations to the node features. We also compare LR-FGDM with the best-performing GAN-based synthetic graph structure generation method, Semantic-aware Node Synthesis (SNS) (Gao et al., 2023b), which is originally proposed to generate synthetic nodes in the minority class for imbalanced datasets with a Generative Adversarial Network (GAN) (Goodfellow et al., 2020). SNS is adapted to the FSNC scenario as a baseline for shot augmentation by generating synthetic support nodes. We further compare LR-FGDM with another shot augmentation method SMILE (Liu et al., 2025b), which augments the support set using a mix-up strategy. For a fair comparison, we apply IA-FSNC, SNS, and SMILE to augment the support set in COLA for 5-way 5-shot FSNC in the same manner as LR-FGDM. It is observed in Table 8 that LR-FGDM always outperforms all the competing shot augmentation methods across multiple graph datasets and few-shot settings.

Table 8: Comparison of LR-FGDM with existing shot augmentation methods on 5-way 5-shot node classification. All methods are applied to augment the COLA (Liu et al., 2024). The statistical significance of the results is deferred to Table 19 of the appendix.

| Method | CoraFull | ogbn-arxiv | Coauthor-CS | DBLP |
|---|---|---|---|---|
| COLA (Liu et al., 2024) | 87.83 | 67.52 | 87.54 | 87.23 |
| COLA (SMILE) (Liu et al., 2025b) | 88.21 | 68.01 | 88.02 | 88.07 |
| COLA (IA-FSNC) (Wu et al., 2022) | 88.36 | 68.17 | 88.21 | 88.16 |
| COLA (SNS) (Gao et al., 2023b) | 88.49 | 68.32 | 88.34 | 88.28 |
| COLA (LR-FGDM) | 89.66 | 69.63 | 89.83 | 89.51 |
| COLA (LRA-LR-FGDM) | **90.32** | **70.22** | **90.39** | **90.07** |

### G.3 STUDY ON THE KERNEL COMPLEXITY (KC) AND THE UPPER BOUND FOR THE TEST LOSS IN THEOREM A.1

In this section, we study the effectiveness of LR-FGDM in reducing the upper bound for the test loss in Equation (5) and the two terms in it, including the kernel complexity of the kernel gram matrix, $KC(\mathbf{K})$, and the training loss of the node classifier with the ground-truth label, $L(\mathbf{K}, \mathbf{Y}_{FS}, t)$. The study is performed for 5-way 5-shot node classification tasks on Cora-Full and Coauthor-CS with two baseline models, COSMIC (Wang et al., 2023a) and COLA (Liu et al., 2024), as well as the corresponding models augmented by LR-FGDM, which are COSMIC (LR-FGDM) and COLA (LR-FGDM). It is observed from Table 9 that the upper bound for the test loss for the few-shot node classifiers trained by LR-FGDM is significantly lower than that of the baseline without low-rank regularization. Furthermore, LR-FGDM achieves substantially lower values in both $KC(\mathbf{K})$ and $L(\mathbf{K}, \mathbf{Y}_{FS}, t)$ in the upper bound, validating the theoretical motivation behind low-rank regularization and confirming the robustness and generalization benefits of LR-FGDM in FSNC.

Table 9: Comparisons on $L(\mathbf{K}, \mathbf{Y}_{FS}, t)$, $KC(\mathbf{K})$, and the upper bound for the test loss. The lowest values for each dataset are bold, and the second-lowest are underlined. The study is performed for 5-way 5-shot FSNC on CoraFull, ogbn-arxiv, Coauthor-CS, DBLP, and Roman-Empire, and for 2-way 5-shot FSNC on Amazon-Computers, Amazon-Photo, and Citeseer.

| Datasets | Terms | COSMIC | COSMIC (LR-FGDM) | COSMIC (LRA-LR-FGDM) | COLA | COLA (LR-FGDM) | COLA (LRA-LR-FGDM) |
|---|---|---|---|---|---|---|---|
| CoraFull | $L(\mathbf{K}, \mathbf{Y}_{FS}, t)$ | 6.44 | 3.72 | 3.57 | 6.38 | 3.65 | **3.52** |
| | KC | 0.35 | 0.20 | 0.19 | 0.40 | 0.18 | **0.17** |
| | Upper Bound | 10.80 | 7.05 | 6.78 | 11.25 | 6.74 | **6.52** |
| ogbn-arxiv | $L(\mathbf{K}, \mathbf{Y}_{FS}, t)$ | 4.54 | 4.02 | 3.89 | 4.69 | 3.95 | **3.82** |
| | KC | 0.47 | 0.24 | 0.23 | 0.50 | 0.21 | **0.20** |
| | Upper Bound | 9.40 | 8.20 | 7.95 | 9.84 | 7.97 | **7.73** |
| Coauthor-CS | $L(\mathbf{K}, \mathbf{Y}_{FS}, t)$ | 4.26 | 3.38 | **3.29** | 3.95 | 3.40 | 3.32 |
| | KC | 0.52 | 0.30 | 0.29 | 0.66 | 0.28 | **0.23** |
| | Upper Bound | 7.99 | 6.25 | 6.09 | 7.63 | 6.16 | **6.03** |
| DBLP | $L(\mathbf{K}, \mathbf{Y}_{FS}, t)$ | 4.63 | 3.75 | 3.63 | 4.41 | 3.58 | **3.46** |
| | KC | 0.48 | 0.26 | 0.25 | 0.53 | 0.23 | **0.21** |
| | Upper Bound | 8.25 | 6.85 | 6.63 | 8.01 | 6.50 | **6.32** |
| Amazon-Computers | $L(\mathbf{K}, \mathbf{Y}_{FS}, t)$ | 5.12 | 3.88 | 3.74 | 4.95 | 3.79 | **3.62** |
| | KC | 0.44 | 0.25 | 0.24 | 0.47 | 0.22 | **0.20** |
| | Upper Bound | 9.12 | 7.10 | 6.93 | 8.80 | 6.85 | **6.60** |
| Amazon-Photo | $L(\mathbf{K}, \mathbf{Y}_{FS}, t)$ | 4.41 | 3.52 | 3.45 | 4.28 | 3.48 | **3.38** |
| | KC | 0.39 | 0.21 | 0.20 | 0.42 | 0.19 | **0.18** |
| | Upper Bound | 8.35 | 6.55 | 6.40 | 8.09 | 6.33 | **6.12** |
| Citeseer | $L(\mathbf{K}, \mathbf{Y}_{FS}, t)$ | 6.92 | 5.10 | 4.98 | 6.78 | 4.95 | **4.82** |
| | KC | 0.58 | 0.31 | 0.29 | 0.61 | 0.27 | **0.22** |
| | Upper Bound | 11.44 | 8.92 | 8.70 | 11.22 | 8.55 | **8.32** |
| Roman-Empire | $L(\mathbf{K}, \mathbf{Y}_{FS}, t)$ | 8.35 | 6.62 | 6.48 | 8.10 | 6.55 | **6.31** |
| | KC | 0.73 | 0.42 | 0.40 | 0.76 | 0.38 | **0.36** |
| | Upper Bound | 13.92 | 11.08 | 10.90 | 13.55 | 10.82 | **10.55** |

### G.4 EFFECTIVENESS OF LR-FGDM ON HETEROPHILIC GRAPHS

While most existing FSNC studies have primarily targeted homophilous graphs, where connected nodes tend to share similar labels, many real-world graphs exhibit heterophily, where neighboring nodes often belong to different classes. In such cases, standard GNN-based few-shot methods face fundamental limitations, as neighborhood aggregation mechanisms become less effective or even detrimental to learning discriminative node representations. This poses an even greater challenge

under few-shot conditions, where only a handful of labeled nodes per class are available to guide the model. To assess the generalization capability of our proposed LR-FGDM in this challenging scenario, we conduct experiments on the Roman-Empire dataset (Platonov et al., 2023) following COLA (Liu et al., 2024), which is a syntactic word dependency graph characterized by extreme heterophily. In this graph, node labels reflect grammatical roles rather than local connectivity, making the graph structure highly non-homophilous. To demonstrate the statistical significance of the improvements achieved by LR-FGDM over the baseline methods, we perform $t$-tests on the few-shot classification accuracies obtained from 20 independent few-shot tasks for each setting. It is observed in Table 11 that models enhanced by LR-FGDM consistently yield statistically significant improvements over the corresponding methods across all few-shot settings with p-values $p < 0.05$.

Table 10: FSNC results on the Roman-Empire dataset, which features extreme heterophily. Accuracy and standard deviation are reported in %. The best result for each setting is in bold, and the second-best is underlined.

| Method | 2-way 1-shot | 2-way 5-shot | 5-way 1-shot | 5-way 5-shot |
|---|---|---|---|---|
| MAML (Finn et al., 2017) | $42.83 \pm 2.31$ | $50.12 \pm 2.24$ | $19.45 \pm 1.20$ | $25.73 \pm 1.41$ |
| ProtoNet (Snell et al., 2017) | $48.67 \pm 2.65$ | $61.34 \pm 2.47$ | $28.52 \pm 1.69$ | $44.10 \pm 1.72$ |
| Meta-GNN (Zhou et al., 2019) | $63.45 \pm 3.20$ | $73.28 \pm 2.85$ | $44.80 \pm 2.14$ | $60.33 \pm 2.06$ |
| GPN (Ding et al., 2020) | $62.10 \pm 3.14$ | $74.01 \pm 2.63$ | $42.73 \pm 2.21$ | $63.45 \pm 1.90$ |
| AMM-GNN (Wang et al., 2020b) | $65.02 \pm 3.05$ | $76.48 \pm 2.13$ | $48.92 \pm 2.39$ | $66.22 \pm 1.84$ |
| G-Meta (Huang & Zitnik, 2020) | $66.74 \pm 3.22$ | $78.36 \pm 2.14$ | $50.14 \pm 2.43$ | $66.40 \pm 1.75$ |
| TENT (Wang et al., 2022) | $66.23 \pm 3.08$ | $77.29 \pm 2.39$ | $45.73 \pm 2.01$ | $61.78 \pm 1.81$ |
| KD-FSNC (Wu et al., 2024) | $69.15 \pm 2.59$ | $80.16 \pm 2.11$ | $58.92 \pm 2.34$ | $73.28 \pm 1.90$ |
| NormProp (Zhang et al., 2025a) | $69.02 \pm 2.67$ | $80.13 \pm 2.14$ | $57.84 \pm 2.30$ | $72.46 \pm 1.92$ |
| COSMIC (Wang et al., 2023a) | $71.84 \pm 2.71$ | $82.35 \pm 2.26$ | $60.25 \pm 2.42$ | $75.33 \pm 2.05$ |
| COLA (Liu et al., 2024) | $70.96 \pm 2.44$ | $81.48 \pm 2.09$ | $59.31 \pm 2.37$ | $74.02 \pm 1.88$ |
| **COSMIC (LR-FGDM)** | $73.38 \pm 2.56$ | $83.79 \pm 2.18$ | $61.45 \pm 2.34$ | $76.52 \pm 1.91$ |
| **COSMIC (LRA-LR-FGDM)** | $73.89 \pm 2.31$ | $84.33 \pm 2.01$ | $61.98 \pm 2.12$ | $77.04 \pm 1.78$ |
| **COLA (LR-FGDM)** | $\underline{75.62 \pm 2.35}$ | $\underline{85.41 \pm 2.02}$ | $\underline{63.18 \pm 2.11}$ | $\underline{78.23 \pm 1.76}$ |
| **COLA (LRA-LR-FGDM)** | $\mathbf{76.17 \pm 2.18}$ | $\mathbf{85.93 \pm 1.89}$ | $\mathbf{63.69 \pm 1.97}$ | $\mathbf{78.79 \pm 1.63}$ |

Table 11: $p$-values from $t$-tests comparing COSMIC (LR-FGDM), COLA (LR-FGDM), COSMIC (LRA-LR-FGDM), and COLA (LRA-LR-FGDM) against their corresponding baseline methods, COSMIC and COLA, on Roman-Empire.

| Task | 2-way 1-shot | 2-way 5-shot | 5-way 1-shot | 5-way 5-shot |
|---|---|---|---|---|
| COSMIC (LR-FGDM) | 0.034 | 0.024 | 0.028 | 0.012 |
| COLA (LR-FGDM) | 0.019 | 0.022 | 0.013 | 0.002 |
| COSMIC (LRA-LR-FGDM) | 0.007 | 0.006 | 0.008 | 0.005 |
| COLA (LRA-LR-FGDM) | 0.004 | 0.003 | 0.006 | 0.003 |

## G.5 CROSS-VALIDATION ON THE NUMBER OF SYNTHETIC NODES

The number of synthetic nodes generated for each novel class given each support node, denoted as $q$, plays a crucial role in determining the effectiveness of LR-FGDM. While generating more synthetic nodes can potentially enrich the support set and provide stronger supervision signals, it may also introduce redundancy or noise if excessive synthetic samples are added. In our experiments, we select the value of $q$ for different datasets using 5-fold cross-validation over the base training classes. The value of $q$ is selected from a range of candidate values, including $\{1, 2, 3, 4, 5, 6, 7, 8, 9, 10\}$.

Table 12: The selected number of synthetic nodes per support node ($q$) for each dataset and few-shot setting.

| Dataset | 2-way 1-shot | 2-way 5-shot | 5-way 1-shot | 5-way 5-shot |
|---|---|---|---|---|
| CoraFull | 2 | 3 | 3 | 5 |
| ogbn-arxiv | 5 | 8 | 5 | 7 |
| Coauthor-CS | 4 | 6 | 4 | 5 |
| DBLP | 5 | 3 | 4 | 5 |

## G.6 SENSITIVITY ANALYSIS ON THE HYPERPARAMETERS $\tau$, $r_0$, $K$, AND $q$

In this section, we first conduct a sensitivity analysis $\tau$, which is the weighting parameter for the TNN $\|\mathbf{K}\|_{r_0}$ in Equation 2. The study is performed using COLA (LR-FGDM) on the CoraFull

dataset for the 5-way 5-shot node classification task. We evaluate the performance of the COLA (LR-FGDM) with $\tau$ varying in $\{0.1, 0.2, 0.3, 0.4, 0.5, 0.6, 0.7, 0.8, 0.9\}$. As shown in Table 13, although the best performance is achieved at $\tau = 0.6$, the performance of COLA (LR-FGDM) remains stable across different values of $\tau$. Even the lowest performing setting, $\tau = 0.1$, results in only a marginal $0.18\%$ decrease in accuracy compared to the best result. In addition, we perform an ablation study to examine the influence of the rank parameter $r_0 = \lceil \gamma \min N, d \rceil$, where $\gamma \in (0, 1]$ controls the effective rank used in the truncated nuclear norm. We evaluate COLA (LR-FGDM) with $\gamma$ varying from 0.05 to 0.5. As shown in Table 13, the accuracy is robust to different values of $\gamma$, with the highest performance observed at $\gamma = 0.2$. We also conduct an ablation study on the hyperparameter $K$, which denotes the number of clusters used for prototype regularization in the HGAE. We vary $K$ from 5 to 50 with a step size of 5. As shown in Table 13, the performance remains stable across different values of $K$, with a slight peak at $K = 10$. This suggests that the model is not sensitive to the choice of cluster number, $K$.

Table 13: Sensitivity analysis on the weighting parameter $\tau$ for the TNN, the rank ratio $\gamma$ used in $r_0 = \lceil \gamma \min\{N, d\} \rceil$, and the number of clusters $K$ for prototype computation in COLA (LR-FGDM) for the 5-way 5-shot node classification task on CoraFull.

| $\tau$ | 0.1 | 0.2 | 0.3 | 0.4 | 0.5 | 0.6 | 0.7 | 0.8 | 0.9 |
|---|---|---|---|---|---|---|---|---|---|
| Accuracy | 89.48 | 89.55 | 89.59 | 89.57 | 89.63 | 89.66 | 89.62 | 89.65 | 89.58 |

| $\gamma$ | 0.05 | 0.1 | 0.15 | 0.2 | 0.25 | 0.3 | 0.35 | 0.4 | 0.45 | 0.5 |
|---|---|---|---|---|---|---|---|---|---|---|
| Accuracy | 89.27 | 89.44 | 89.58 | 89.66 | 89.62 | 89.55 | 89.64 | 89.60 | 89.60 | 89.55 |

| $K$ | 5 | 10 | 15 | 20 | 25 | 30 | 35 | 40 | 45 | 50 |
|---|---|---|---|---|---|---|---|---|---|---|
| Accuracy | 89.30 | 89.66 | 89.58 | 89.64 | 89.53 | 89.59 | 89.62 | 89.56 | 89.48 | 89.44 |

Furthermore, we perform a sensitivity analysis to evaluate the effect of the number of synthetic nodes per support node, $q$, on FSNC performance under the 5-way 5-shot setting. For each dataset, we vary $q$ from 1 to 10 and report the mean classification accuracy over 20 tasks. As shown in Table 14, increasing $q$ generally leads to consistent improvements in performance up to a certain threshold, beyond which the gains tend to saturate or marginally decline. This trend highlights the benefit of augmenting the support set with a moderate number of synthetic nodes, which helps enhance generalization by enriching the local representation space. Notably, while further increasing $q$ beyond the optimal value does not continue to improve performance, the resulting degradation is marginal.

Table 14: Sensitivity analysis on the number of synthetic nodes per support node ($q$) for four datasets under the 5-way 5-shot FSNC setting.

| $q$ | 1 | 2 | 3 | 4 | 5 | 6 | 7 | 8 | 9 | 10 |
|---|---|---|---|---|---|---|---|---|---|---|
| CoraFull | 88.71 | 89.02 | 89.37 | 89.51 | 89.66 | 89.53 | 89.59 | 89.44 | 89.41 | 89.40 |
| ogbn-arxiv | 67.80 | 68.21 | 68.84 | 69.08 | 69.33 | 69.42 | 69.63 | 69.59 | 69.45 | 69.20 |
| Coauthor-CS | 88.33 | 88.82 | 89.08 | 89.31 | 89.83 | 89.62 | 89.54 | 89.48 | 89.30 | 89.12 |
| DBLP | 87.93 | 88.44 | 88.87 | 89.02 | 89.51 | 89.35 | 89.29 | 89.10 | 88.94 | 88.91 |

## G.7 QUALITY EVALUATION OF THE AUGMENTED GRAPH

This paper introduces a novel node-level graph diffusion model named FGDM, which synthesizes the graph structures. The synthetic graph structure, which consists of the synthetic support nodes and the associated edges, generated by the FGDM, is subsequently combined with the original graph to form an augmented graph. In Section 4.2, we have shown that the FSNC methods trained on the augmented graph achieve significantly better performance. In this section, we directly evaluate the data quality of the synthetic graph structures generated by the FGDM. In the visual domain, the Frechet Inception Distance (FID) is a widely used metric to evaluate the quality of the synthetic images generated by the generative models (Brock et al., 2019; Ho et al., 2020). The FID score measures the similarity between the distribution of the generated images and the distribution of the real images. To compute the FID score, the pre-trained Inception v3 (Szegedy et al., 2016) is used to extract the features from both the real images and the generated images, which are then modeled as the multivariate Gaussian distributions. The FID score is then calculated using the Frechet Distance

(FD) (Brock et al., 2019) between the two multivariate Gaussian distributions modeling the real and the generated images (Dowson & Landau, 1982). A lower FID score indicates that the generated images are more similar to the real images, suggesting better quality.

**Quality Evaluation of the Synthetic Nodes.** Although the Inception model cannot be applied to the graph data, we can replace the Inception model in the computation of the FID score with the pre-trained GCN (Kipf & Welling, 2017) for extracting node features to adapt the metric to evaluate the quality of synthetic nodes generated by the FGDM. To this end, we define the Frechet Node Distance (FND), which is the FD between the multivariate Gaussians modeling the node features extracted by pre-trained GCN. We randomly split the nodes from the novel classes in the original graph into two partitions of equal size, which are the base partition and the test partition. To mitigate the influence of the randomness, we compute the FND scores with 10 different random splits and report the mean and the standard deviation of the FND scores across different runs. The FND computed between the nodes in the test partition and the base partition of the original graph establishes the baseline of the expected FND score for high-quality support nodes. By computing the FND score between the features of the synthetic support nodes in the synthetic graph structures generated by the FGDM and the features of the nodes in the base partition of the original graph, we evaluate the quality of the synthetic support nodes. For simplicity, we refer to the FND score for the synthetic support nodes as the FND between their features and the features of the nodes in the base partition of the original graph. To show the effectiveness of the prototypical regularization in the training of the HGAE for the PGDM, we also compute the FND for the nodes in the synthetic graph structures generated by the PGDM without the prototypical regularization. To demonstrate the advantages of the FGDM over the vanilla diffusion model, the DDPM (Ho et al., 2020), we train a baseline DDPM model on the input node attributes of the original graph and synthesize the same number of synthetic nodes as the FGDM. The synthetic edges are then generated by connecting each synthetic node to its K-nearest neighbors in the original graph using the K-nearest neighbors (KNN) algorithm with $K = \lceil d_{ave} \rceil$, where $d_{ave}$ is the average degree of the original graph. The synthetic graph structures, including the synthetic nodes and edges generated by the baseline DDPM model, are combined with the original graph to form the augmented graph. Next, we compute the FND score for the nodes in the synthetic graph structures generated by the baseline DDPM model. In addition, we also compute the FND for the support nodes generated by three competing shot augmentation methods, including SMILE (Liu et al., 2025b), IA-FSNC (Wu et al., 2022), and SNS (Gao et al., 2023b). The ablation study is performed for the 5-way 5-shot setting of the FSNC. The lower FND scores indicate that the node features are more similar to the features of nodes in the base partition of the original graph. It is observed in Table 15 that the FND score of the nodes in the synthetic graph structures generated by the FGDM is closest to the FND score of the original graph, which demonstrates that the FGDM generates faithful synthetic nodes.

Table 15: Frechet Node Distance (FND) to the nodes in the base partition of the original graph. The mean and standard deviation of the FND scores computed with 10 different random splits of the base partition and the test partition in the original graph are reported. The evaluation is performed for the 5-way 5-shot setting of the FSNC. The FND score for the original graph is computed between the nodes in the test partition and the nodes in the base partition of the original graph.

| Data | CoraFull | ogbn-arxiv | Coauthor-CS | DBLP |
|---|---|---|---|---|
| Baseline DDPM | 13.41±0.43 | 8.95±0.28 | 10.32±0.41 | 7.34±0.39 |
| SMILE (Liu et al., 2025b) | 10.21±0.39 | 6.03±0.32 | 8.21±0.43 | 6.48±0.34 |
| IA-FSNC (Wu et al., 2022) | 9.92±0.41 | 5.87±0.29 | 7.88±0.36 | 6.12±0.37 |
| SNS (Gao et al., 2023b) | 9.73±0.40 | 5.69±0.31 | 7.45±0.34 | 6.01±0.33 |
| FGDM (w/o Prototypical Regularization) | 9.24±0.48 | 5.23±0.30 | 7.13±0.47 | 5.95±0.36 |
| FGDM | **8.10±0.27** | **4.39±0.30** | **5.29±0.21** | **4.08±0.42** |
| Original Graph | 7.95±0.32 | 4.33±0.27 | 4.21±0.26 | 3.84±0.33 |

**Quality Evaluation of the Synthetic Edges.** Similar to the design of the FND score for evaluating the quality of synthetic nodes, we replace the Inception model in the computation of FID with the pre-trained GNN (Zhu et al., 2021) for edge feature extraction to adapt the metric to evaluate the quality of the synthetic edges generated by the FGDM. To this end, we define Frechet Edge Distance (FED), which is the FD between the multivariate Gaussians modeling the edge features extracted by a pre-trained GNN. Similar to the evaluation of the FND, we randomly split the edges in the original graph into two partitions of equal size, which are the base partition and the test par-

tition. To mitigate the influence of the randomness, we compute the FED scores with 10 different random splits and report the mean and the standard deviation of the FED across different runs. The FED computed between the edges in the test partition and the edges in the base partition of the original graph establishes the baseline of the expected FED score for the high-quality edges. By computing the FED between the features of edges in the synthetic graph structures generated by the FGDM and the features of edges in the base partition of the original graph, we evaluate the quality of the edges in the synthetic graph structures. For simplicity, we refer to the FED score for the synthetic edges as the FED between their features and the features of edges in the base partition of the original graph. Similar to the evaluation of the synthetic nodes, we also compute the FED score for the edges in the synthetic graph structures generated by the FGDM without the prototypical regularization. We compute the FED score for the edges in the synthetic graph structures generated by the baseline DDPM model. Since the edges in the synthetic graph structures are generated by the KNN algorithm, we evaluate the baseline DDPM models using different values of $K$ from $\{\lceil d_{\mathrm{ave}}/4 \rceil, \lceil d_{\mathrm{ave}}/2 \rceil, \lceil d_{\mathrm{ave}} \rceil, 2 \times \lceil d_{\mathrm{ave}} \rceil, 4 \times \lceil d_{\mathrm{ave}}/4 \rceil \}$. In addition, we also compute the FED for the edges generated by three competing shot augmentation methods, including SMILE (Liu et al., 2025b), IA-FSNC (Wu et al., 2022), and SNS (Gao et al., 2023b). The FED score for edges in the original graph is also computed. We use the same NBFNet (Zhu et al., 2021) pre-trained on the original graph to extract the edge features for computing the FED score. The ablation study is performed for the 5-way 5-shot setting of the FSNC. Lower FED scores indicate that the edge features are more similar to the features of edges in the base partition of the original graph. It is observed in Table 16 that the FED score of the edges in the synthetic graph structures generated by the FGDM is closest to the FED score of the original graph, which demonstrates that the FGDM generates more faithful synthetic edges compared to the competing methods.

Table 16: Frechet Edge Distance (FED) to the edges in the base partition of the original graph. The mean and standard deviation of the FED scores computed with 10 different random splits of the base partition and the test partition in the original graph are reported. The evaluation is performed for the 5-way 5-shot setting of the FSNC. The FED score for the original graph is computed between the nodes in the test partition and the nodes in the base partition of the original graph.

| Data | CoraFull | ogbn-arxiv | Coauthor-CS | DBLP |
|---|---|---|---|---|
| Baseline DDPM ($K = \lceil d_{\mathrm{ave}}/4 \rceil$) | 12.17±0.47 | 8.85±0.30 | 10.89±0.38 | 8.05±0.41 |
| Baseline DDPM ($K = \lceil d_{\mathrm{ave}}/2 \rceil$) | 11.34±0.36 | 8.53±0.28 | 9.43±0.34 | 7.32±0.33 |
| Baseline DDPM ($K = \lceil d_{\mathrm{ave}} \rceil$) | 10.48±0.39 | 8.01±0.31 | 9.04±0.36 | 6.88±0.32 |
| Baseline DDPM ($K = \lceil 2 \times d_{\mathrm{ave}} \rceil$) | 10.51±0.35 | 7.96±0.29 | 9.17±0.37 | 6.94±0.34 |
| Baseline DDPM ($K = \lceil 4 \times d_{\mathrm{ave}} \rceil$) | 10.92±0.42 | 8.05±0.33 | 9.41±0.39 | 7.10±0.36 |
| SMILE (Liu et al., 2025b) | 10.41±0.36 | 7.89±0.30 | 8.93±0.34 | 6.67±0.31 |
| IA-FSNC (Wu et al., 2022) | 10.96±0.34 | 7.80±0.29 | 8.62±0.33 | 6.39±0.30 |
| SNS (Gao et al., 2023b) | 10.18±0.35 | 7.77±0.28 | 8.66±0.32 | 6.46±0.29 |
| FGDM (w/o Prototypical Regularization) | 10.13±0.38 | 7.70±0.27 | 8.79±0.35 | 6.33±0.30 |
| FGDM | **8.29±0.30** | **5.33±0.24** | **6.56±0.26** | **5.39±0.27** |
| Original Graph | 8.10±0.27 | 5.14±0.22 | 6.37±0.25 | 5.19±0.26 |

Table 17: $p$-values from $t$-tests comparing COSMIC (LR-FGDM), COLA (LR-FGDM), COSMIC (LRA-LR-FGDM), and COLA (LRA-LR-FGDM) against their corresponding baseline methods, COSMIC and COLA, in Table 1.

| Dataset | CoraFull | | | | ogbn-arxiv | | | |
|---|---|---|---|---|---|---|---|---|
| Task | 2-way 1-shot | 2-way 5-shot | 5-way 1-shot | 5-way 5-shot | 2-way 1-shot | 2-way 5-shot | 5-way 1-shot | 5-way 5-shot |
| COSMIC (LR-FGDM) | 0.031 | 0.022 | 0.018 | 0.014 | 0.027 | 0.039 | 0.044 | 0.017 |
| COLA (LR-FGDM) | 0.009 | 0.005 | 0.006 | 0.004 | 0.012 | 0.007 | 0.011 | 0.003 |
| COSMIC (LRA-LR-FGDM) | 0.007 | 0.005 | 0.008 | 0.006 | 0.004 | 0.003 | 0.007 | 0.005 |
| COLA (LRA-LR-FGDM) | 0.004 | 0.003 | 0.007 | 0.005 | 0.006 | 0.004 | 0.008 | 0.002 |

| Dataset | Coauthor-CS | | | | DBLP | | | |
|---|---|---|---|---|---|---|---|---|
| Task | 2-way 1-shot | 2-way 5-shot | 5-way 1-shot | 5-way 5-shot | 2-way 1-shot | 2-way 5-shot | 5-way 1-shot | 5-way 5-shot |
| COSMIC (LR-FGDM) | 0.013 | 0.015 | 0.034 | 0.025 | 0.019 | 0.011 | 0.027 | 0.014 |
| COLA (LR-FGDM) | 0.007 | 0.003 | 0.009 | 0.004 | 0.006 | 0.002 | 0.005 | 0.003 |
| COSMIC (LRA-LR-FGDM) | 0.006 | 0.004 | 0.008 | 0.007 | 0.005 | 0.004 | 0.009 | 0.006 |
| COLA (LRA-LR-FGDM) | 0.003 | 0.002 | 0.006 | 0.005 | 0.004 | 0.001 | 0.007 | 0.003 |

## G.8 IMPROVEMENT STATISTICAL SIGNIFICANCE ANALYSIS

To demonstrate the statistical significance of the improvements achieved by LR-FGDM over the baseline methods, we perform $t$-tests on the few-shot classification accuracies obtained from 20 in-

Table 18: $p$-values from $t$-tests comparing COLA (LR-FGDM) and COLA (LRA-LR-FGDM) with the second-best ablation model in Table 2.

| Dataset | CoraFull | ogbn-arxiv | Coauthor-CS | DBLP |
|---|---|---|---|---|
| COLA (LR-FGDM) | 0.012 | 0.009 | 0.043 | 0.008 |
| COLA (LRA-LR-FGDM) | 0.004 | 0.003 | 0.007 | 0.004 |

Table 19: $p$-values from $t$-tests comparing COLA (LR-FGDM) and COLA (LRA-LR-FGDM) with the second-best ablation model in Table 8.

| Dataset | CoraFull | ogbn-arxiv | Coauthor-CS | DBLP |
|---|---|---|---|---|
| COLA (LR-FGDM) | 0.025 | 0.032 | 0.026 | 0.018 |
| COLA (LRA-LR-FGDM) | 0.005 | 0.004 | 0.006 | 0.003 |

dependent few-shot tasks for each setting. For results in Table 1, we compare COSMIC (LR-FGDM) and COLA (LR-FGDM) against their corresponding baseline methods, COSMIC and COLA. It is observed in Table 17 that models enhanced by LR-FGDM consistently yield statistically significant improvements over the corresponding methods across all datasets and settings with p-values $p < 0.05$. In addition, we further validate the statistical significance of the improvements of COLA (LR-FGDM) over the ablation models in Table 2 and models augmented by other support set augmentation methods in Table 8. It is observed from Table 18 and Table 19 that COLA (LR-FGDM) significantly outperforms all competing variants, with all $p$-values below 0.05, further supporting the statistical significance of the improvements by LR-FGDM over existing methods.

### G.9 EIGEN-PROJECTION AND CONCENTRATION RATIO ANALYSIS

We first compute the eigenvectors $\mathbf{U}$ of the feature gram matrix $\mathbf{K}$. Let $\mathbf{U}^{(1:r)} \in \mathbb{R}^{N \times r}$ be the top $r$-eigenvectors of $\mathbf{K}$ and $\mathbf{U}^{(r)}$ be the $r$-th eigenvector of $\mathbf{K}$. Then, the eigen-projection value of the ground-truth label $\mathbf{Y}_{\text{FS}}$ on $\mathbf{U}^{(r)}$ is computed by $p_r = \frac{1}{n} \sum_{c=1}^{n} \left\| \mathbf{U}^{(r)\top} \mathbf{Y}_{\text{FS}}^{(c)} \right\|_2^2 / \left\| \mathbf{Y}_{\text{FS}}^{(c)} \right\|_2^2$ for $r \in [N]$, where $\mathbf{Y}_{\text{FS}}^{(c)}$ is the $c$-th column of $\mathbf{Y}_{\text{FS}}$. We let $\mathbf{p} = [p_1, \ldots, p_N] \in \mathbb{R}^N$. The eigen-projection $p_r$ reflects the amount of the signal in the label projected onto the $r$-th eigenvector of $\mathbf{K}$, and the signal concentration ratio of a rank $r$ reflects the proportion of signal projected onto the top $r$ eigenvectors of $\mathbf{K}$. The signal concentration ratio for rank $r$ is computed by $\|\mathbf{p}(1:r)\|_1$, where $\mathbf{p}(1:r)$ contains the first $r$ elements of $\mathbf{p}$. Figure 6 illustrates the eigen-projection and concentration ratio of the ground truth label on Cora-Full. By the rank $r = r_0 = 0.2 \min\{N, D'\}$, the signal concentration ratio of the ground truth label is 0.792.

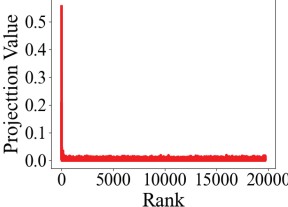 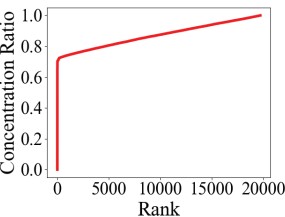

Figure 6: Eigen-projection (left) and concentration ratio (right) of the ground truth label on Cora-Full.

### G.10 STUDY ON THE AUTOMATIC SELECTION OF THE RANK $r_0$

We explore an automatic rank-selection strategy based on the eigenvalue decay of the feature kernel Gram matrix $\mathbf{K}$. In particular, we evaluate an eigengap-based method widely used in spectral dimensionality selection, in which $r_0$ is selected at the index corresponding to the largest relative decay between consecutive eigenvalues (Ng et al., 2001). As a result, $r_0$ is computed as $r_0 = \arg\max_{1 \leq i < N} \frac{\widehat{\lambda}_i - \widehat{\lambda}_{i+1}}{\widehat{\lambda}_i}$. The study is performed using the base FSNC model, COLA (Liu et al., 2024). As shown in Table 20, the performance of LR-FGDM and LRA-LR-FGDM with the automatically selected rank $r_0$ is only marginally different from that achieved via cross-validation.

The deviations remain consistently marginal, typically smaller than $0.2\%$. The results suggest that the automatic rank-selection strategy based on the eigenvalue decay rate provides an efficient alternative to tuning the rank $r_0$, without sacrificing performance.

Table 20: Ablation study on the selection strategy of $r_0$ for the low-rank regularization within LR-FGDM and LRA-LR-FGDM. Results are reported under the 5-way 5-shot FSNC setting.

| Method | Selection Method of $r_0$ | CoraFull | ogbn-arxiv | Coauthor-CS | DBLP |
|---|---|---|---|---|---|
| COLA (Liu et al., 2024) | - | 87.83 | 67.52 | 87.54 | 87.23 |
| COLA (LR-FGDM) | Cross-Validation | 89.66 | 69.63 | 89.83 | 89.51 |
| COLA (LR-FGDM) | Automatic | 89.56 | 69.68 | 89.74 | 89.55 |
| COLA (LRA-LR-FGDM) | Cross-Validation | 90.32 | 70.22 | 90.39 | 90.07 |
| COLA (LRA-LR-FGDM) | Automatic | 90.36 | 70.04 | 90.21 | 90.11 |

### G.11 T-SNE VISUALIZATION COMPARING THE SYNTHETIC NODES AND REAL NODES IN THE NOVEL CLASSES

To provide a more intuitive assessment of the semantic correctness of the generated nodes, we perform a t-SNE visualization study in this section. The visualization results illustrate the spatial distribution of the node embeddings of the synthetic nodes generated by LR-FGDM compared to the node embeddings of the real nodes in the corresponding few-shot classes. For each of the 5-way 5-shot novel classes, 20 real nodes are randomly sampled, and 20 synthetic nodes are generated by FGDM for comparison. As illustrated in Figure 7, embeddings of the synthetic nodes produced by LR-FGDM form compact clusters aligned closely with the real node clusters, demonstrating that the generated nodes preserve both semantic structure and class-level discriminativeness. These qualitative results complement the quantitative FND/FED metrics and provide additional evidence that LR-FGDM generates semantically meaningful node representations.

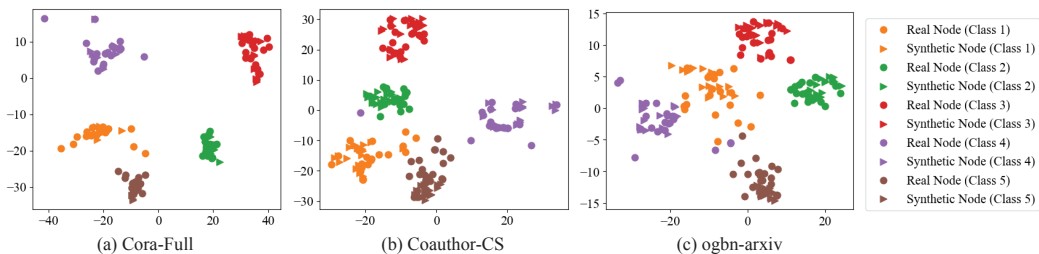

(a) Cora-Full      (b) Coauthor-CS      (c) ogbn-arxiv

Figure 7: t-SNE visualization of the embeddings of real nodes and synthetic nodes generated by FGDM. For each of the 5-way 5-shot novel classes, 20 real nodes are randomly sampled, and 20 synthetic nodes are generated by FGDM for comparison.

### G.12 ABLATION STUDY ON THE LDM

To demonstrate the contribution of the proposed generative mechanism, we perform an additional ablation study in which the LDM in LR-FGDM is replaced with two alternative generative models, including a Variational Autoencoder (VAE) (Kipf & Welling, 2016a) and a generic diffusion model, DDPM (Ho et al., 2020). The study is performed using the base FSNC model, COLA (Liu et al., 2024). The ablation models are denoted as COLA (DDPM) and COLA (VAE). It is observed in Table 21 that COLA (LR-FGDM) significantly outperforms the two ablation models, COLA (DDPM) and COLA (VAE), demonstrating the better quality of the augmented graphs generated by FGDM.

Table 21: Ablation study on the generative model used in LR-FGDM. Results are reported under the 5-way 5-shot FSNC setting.

| Method | CoraFull | ogbn-arxiv | Coauthor-CS | DBLP |
|---|---|---|---|---|
| COLA (Liu et al., 2024) | 87.83 | 67.52 | 87.54 | 87.23 |
| COLA (DDPM) | 88.85 | 68.35 | 88.42 | 88.12 |
| COLA (VAE) | 88.47 | 68.26 | 88.30 | 88.07 |
| COLA (LR-FGDM) | 89.56 | 69.68 | 89.74 | 89.55 |

### G.13 TRAINING TIME COMPARISON AGAINST BASELINE FSNC METHODS

We report the training time of LR-FGDM and LRA-FGDM when integrated with COSMIC and COLA on the augmented graphs in Table 22. The evaluation is performed on a single 80G NVIDIA A100 GPU. It is observed that LR-FGDM introduces additional computational overhead due to the training of the HGAE, the LDM, and the subsequent low-rank transductive classifier. However, the increase in training time is moderate relative to the performance improvements it delivers. While lightweight metric-based methods such as ProtoNet and NormProp remain the fastest, their accuracy is substantially lower across all benchmarks. In contrast, strong meta-learning and contrastive methods already incur significantly higher training costs. LR-FGDM adds only a modest overhead on top of COSMIC and COLA while consistently improving accuracy on all datasets. Across all datasets, the increase in training time introduced by LR-FGDM and LRA-LR-FGDM is in the range of 3.4%–6.2% compared to the corresponding baselines, COSMIC and COLA.

Table 22: Training time comparison (in seconds) and corresponding 5-way 5-shot accuracy (%) under the 5-way 5-shot setting. The training time for the HGAE and LDM in FGDM and the generation time of the synthetic graph structures by FGDM are reported in Table 3.

| Method | CoraFull | | Coauthor-CS | | ogbn-arxiv | | DBLP | |
|---|---|---|---|---|---|---|---|---|
| | Time | Acc | Time | Acc | Time | Acc | Time | Acc |
| ProtoNet | 19.4 | 51.54% | 15.8 | 49.25% | 40.6 | 53.31% | 22.9 | 52.26% |
| Meta-GNN | 83.3 | 70.50% | 68.7 | 68.59% | 125.1 | 31.52% | 92.4 | 72.15% |
| GPN | 53.0 | 72.82% | 42.4 | 81.79% | 171.8 | 50.50% | 63.7 | 75.46% |
| G-Meta | 662.5 | 75.84% | 542.1 | 74.18% | 598.4 | 47.16% | 328.9 | 75.82% |
| TENT | 58.9 | 70.10% | 47.8 | 76.90% | 84.2 | 61.38% | 71.6 | 74.84% |
| KD-FSNC | 74.6 | 85.89% | 60.8 | 84.42% | 112.3 | 64.91% | 83.5 | 83.75% |
| NormProp | 22.3 | 85.47% | 18.1 | 84.61% | 41.9 | 64.28% | 25.8 | 84.12% |
| STAR | 267.4 | 87.31% | 217.6 | 87.60% | 398.5 | 66.98% | 302.7 | 87.10% |
| DoG | 850.5 | 86.47% | 690.2 | 87.35% | 1225.3 | 65.69% | 1042.4 | 87.59% |
| COSMIC | 824.9 | 86.34% | 668.2 | 85.47% | 1221.7 | 65.42% | 1015.3 | 85.31% |
| COLA | 781.6 | 87.83% | 632.3 | 87.54% | 1155.1 | 67.52% | 958.6 | 87.23% |
| COSMIC (LR-FGDM) | 854.2 | 88.81% | 691.3 | 87.72% | 1266.5 | 67.59% | 1056.3 | 87.44% |
| COSMIC (LRA-LR-FGDM) | 875.3 | 89.47% | 704.4 | 88.21% | 1295.1 | 68.11% | 1078.1 | 87.96% |
| COLA (LR-FGDM) | 810.5 | 89.66% | 652.3 | 89.83% | 1200.2 | 69.63% | 992.4 | 89.51% |
| COLA (LRA-LR-FGDM) | 829.2 | 90.32% | 670.4 | 90.39% | 1221.5 | 70.22% | 1008.2 | 90.07% |

### G.14 STRUCTURAL PROPERTY EVALUATION OF THE AUGMENTED GRAPH

To demonstrate that the synthetic edges preserve key graph properties, we evaluate two quantitative graph structural properties, including the small-world property (Telesford et al., 2011) and the power-law degree distribution property (Clauset et al., 2009; Zhao et al., 2024b). The small-world property is measured by the small-world coefficient $\sigma = \frac{C/C_{\mathrm{rand}}}{L/L_{\mathrm{rand}}}$ where $C$ and $L$ denote the clustering coefficient and characteristic path length of the augmented graph, and $C_{\mathrm{rand}}, L_{\mathrm{rand}}$ are the clustering coefficient and characteristic path length a degree-preserving random graph. The power-law degree distribution property is measured by the power-law exponent and the Kolmogorov–Smirnov (KS) statistic between the empirical degree distribution and a fitted power-law. To evaluate the two structural properties above of the augmented graphs obtained by LR-FGDM, we compute the small-world coefficient $\sigma$ and the KS statistic for both the edges in the original graphs and their LR-FGDM–augmented counterparts across CoraFull, ogbn-arxiv, Coauthor-CS, and DBLP. The study is performed for the 5-way 5-shot FSNC setting. It is observed in Table 23 that the synthetic graph structures generated by LR-FGDM exhibit a small-world coefficient $\sigma$ and power-law exponent $\alpha$ that closely match those of the original graph, with only marginal deviation in the KS statistic. These results confirm that the hierarchical edge reconstruction in HGAE successfully preserves both the small-world characteristics and the heavy-tailed degree patterns of real-world graphs.

## H ALGORITHMS FOR THE TRAINING OF LR-FGDM

We present the training algorithm of the FGDM in Algorithm 1, which comprises two steps. The first step, which is from Line 1 to Line 5 in Algorithm 1, describes the training of the HGAE. The second

Table 23: Structural property evaluation of the augmented graph. We report the small-world coefficient $\sigma$ and KS statistic for both the original graph and the LR-FGDM–augmented graph.

| Metric | Graph | CoraFull | ogbn-arxiv | Coauthor-CS | DBLP |
|--------|-------|----------|------------|-------------|------|
| $\sigma$ | Original | 3.27 | 2.56 | 4.11 | 2.99 |
| | Augmented | 3.12 | 2.48 | 3.99 | 2.94 |
| KS | Original | 0.067 | 0.054 | 0.062 | 0.049 |
| | Augmented | 0.068 | 0.056 | 0.061 | 0.052 |

---

**Algorithm 1** Training FGDM (Training the HGAE and the LDM)

---

**Input:** The input attribute matrix $\mathbf{X}$, adjacency matrix $\mathbf{A}$, the training epochs of the HGAE $t_{\text{HGAE}}$, the labels $Y_{\text{base}}$ of the labeled nodes $\mathcal{V}_{\text{base}}$ in the base training set, the training epochs of the LDM $t_{\text{LDM}}$, and the learning rate $\eta$

**Output:** The parameters of the HGAE $\boldsymbol{\omega}$ and the parameters of the LDM $\boldsymbol{\theta}$

1: Obtain the inter-cluster neighbor map $\mathbf{C}$ and the intra-cluster neighbor map $\mathbf{M}$ by applying balanced $K$-Means clustering on $\mathbf{X}$
2: Initialize the parameter $\boldsymbol{\omega}$ of the HGAE
3: **for** $t \leftarrow 1$ to $t_{\text{HGAE}}$ **do**
4:      Update $\boldsymbol{\omega}$ by $\boldsymbol{\omega} \leftarrow \boldsymbol{\omega} - \eta \nabla_{\boldsymbol{\omega}} L_{\text{HGAE}}$ with $L_{\text{HGAE}}$ from Eq.(1)
5: **end for**
6: Compute cluster prototypes $\{\mathbf{p}_c\}$ from the latent representations $\mathbf{Z}$ using semi-supervised K-means clustering (Bair, 2013)
7: Assign each node to its cluster prototype based on the clustering result
8: Initialize the parameter $\boldsymbol{\theta}$ of the LDM
9: Map the node attributes $\mathbf{X}$ and the adjacency matrix $\mathbf{A}$ to the latent space using the encoder $g_e$ of the HGAE as $\mathbf{H} = g_e(\mathbf{X}, \mathbf{A})$
10: Train the LDM with CFG (Ho & Salimans, 2022) on latent pairs $(\mathbf{Z}_i, \mathbf{p}_{\pi(i)})$ where $\mathbf{p}_{\pi(i)}$ is the prototype associated with node $v_i$
11: **return** The parameters of the HGAE $\boldsymbol{\omega}$ and the parameters of the LDM $\boldsymbol{\theta}$

---

**Algorithm 2** Generation of the Augmented Graph $\mathcal{G}_{\text{aug}}$

---

**Input:** The input attribute matrix $\mathbf{X}$, the adjacency matrix $\mathbf{A}$, the set of support nodes $\mathcal{V}_{\text{sup}} = \{v_1, \ldots, v_S\}$, number of synthetic nodes per support node $q$, and the prototype assignments $\{\mathbf{p}_{\pi(i)}\}_{i=1}^{S}$

**Output:** The augmented graph $\mathcal{G}_{\text{aug}} = (\mathcal{V} \cup \mathcal{V}_{\text{syn}}, \mathbf{X}_{\text{aug}}, \mathbf{A}_{\text{aug}})$

1: Let $M = q \times |\mathcal{V}_{\text{sup}}|$, total number of synthetic nodes
2: Initialize counter $m \leftarrow 1$
3: **for** $i \leftarrow 1$ to $|\mathcal{V}_{\text{sup}}|$ **do**
4:      **for** $j \leftarrow 1$ to $q$ **do**
5:          Sample noise $\boldsymbol{\varepsilon} \sim \mathcal{N}(\mathbf{0}, \mathbf{I})$
6:          Condition on prototype $\mathbf{p}_{\pi(i)}$ of support node $v_i$
7:          Generate latent feature $\widehat{\mathbf{Z}}_m$ using LDM conditioned on $\mathbf{p}_{\pi(i)}$
8:          $m \leftarrow m + 1$
9:      **end for**
10: **end for**
11: Decode $\widehat{\mathbf{Z}} = \{\widehat{\mathbf{Z}}_i\}_{i=1}^{M}$ to $\mathbf{X}_{\text{syn}}$ and $\mathbf{A}_{\text{syn}}$ with the decoder of the HGAE $(\mathbf{X}_{\text{syn}}, \mathbf{A}_{\text{syn}}) = g_d(\widehat{\mathbf{Z}})$
12: Form augmented attribute matrix: $\mathbf{X}_{\text{aug}} = [\mathbf{X}; \mathbf{X}_{\text{syn}}]$
13: Form augmented adjacency matrix: $\mathbf{A}_{\text{aug}} = [\mathbf{A} \; \mathbf{A}_{\text{syn}}; \mathbf{A}_{\text{syn}} \; \mathbf{A}]$
14: **return** $\mathcal{G}_{\text{aug}} = (\mathcal{V} \cup \mathcal{V}_{\text{syn}}, \mathbf{X}_{\text{aug}}, \mathbf{A}_{\text{aug}})$

---

step, which is from Line 6 to Line 10, describes the training of the LDM. Algorithm 2 describes the generation process for the augmented graph $\mathcal{G}_{\text{aug}}$.

## I   Efficient Computation of the TNN $\|\mathbf{K}\|_{r_0}$

To approximate the top-$r_0$ eigenvectors, $\mathbf{U}^{(r_0)} \in \mathbb{R}^{N \times r_0}$, of the gram matrix $\mathbf{K}$ using the Nystrom method (Kumar et al., 2012), we first sample $M'$ landmark points from the training set, indexed by $\mathcal{I} \subset [N]$ with $|\mathcal{I}| = M' \ll N$. Let $\mathbf{F}_{\mathcal{I}} \in \mathbb{R}^{M' \times d}$ be the features corresponding to the landmark

set. We define $\mathbf{C} = \mathbf{F}\mathbf{F}_{\mathcal{I}}^{\top} \in \mathbb{R}^{N \times M'}$ as the cross-covariance matrix and $\mathcal{W} = \mathbf{F}_{\mathcal{I}}\mathbf{F}_{\mathcal{I}}^{\top} \in \mathbb{R}^{M' \times M'}$ as the gram matrix on the landmarks. Next, we compute the top-$r_0$ eigen-decomposition of $\mathcal{W}$ as $\mathcal{W} = \mathbf{Q}\mathbf{\Lambda}\mathbf{Q}^{\top}$, where $\mathbf{Q} \in \mathbb{R}^{M' \times r_0}$ contains the top-$r_0$ eigenvectors and $\mathbf{\Lambda} \in \mathbb{R}^{r_0 \times r_0}$ is the diagonal matrix of corresponding eigenvalues. The Nystrom approximation of $\mathbf{K}$ is then given by $\tilde{\mathbf{K}} = \mathbf{C}\mathcal{W}^{\dagger}\mathbf{C}^{\top}$, and the approximate top-$r_0$ eigenvectors are computed as $\tilde{\mathbf{U}}^{(r_0)} = \mathbf{C}\mathbf{Q}\mathbf{\Lambda}^{-1/2} \in \mathbb{R}^{N \times r_0}$, which serves as an efficient approximation to $\mathbf{U}^{(r_0)}$ with significantly reduced computational cost.

We let $\mathbf{U}_{r_0} = \tilde{\mathbf{U}}^{(r_0)}$, then the sum of the top-$r_0$ eigenvalues of $\mathbf{K}$ is approximated by $\mathrm{tr}\left(\mathbf{U}_{r_0}^{\top}\mathbf{K}\mathbf{U}_{r_0}\right)$. Since $\mathrm{tr}\left(\mathbf{K}\right) = \sum_{i=1}^{N} K(\mathbf{x}_i, \mathbf{x}_i) = \sum_{i=1}^{N} \widehat{\lambda}_i$, and $\mathrm{tr}\left(\mathbf{U}_{r_0}^{\top}\mathbf{K}\mathbf{U}_{r_0}\right) = \sum_{i=1}^{N}(\sum_{s=1}^{r}\sum_{k=1}^{N}[\mathbf{U}_{r_0}]_{si}^{\top}[\mathbf{K}]_{ik}[\mathbf{U}_{r_0}]_{ks})$, we can approximate the TNN $\|\mathbf{K}\|_{r_0}$ by $\overline{\|\mathbf{K}\|_{r_0}} = \mathrm{tr}\left(\mathbf{K}\right) - \mathrm{tr}\left(\mathbf{U}_{r_0}^{\top}\mathbf{K}\mathbf{U}_{r_0}\right)$ which is separable. In particular, $\overline{\|\mathbf{K}\|_{r_0}} = \sum_{i=1}^{N} K(\mathbf{x}_i, \mathbf{x}_i) - \sum_{i=1}^{N}\left(\sum_{s=1}^{r}\sum_{k=1}^{N}[\mathbf{U}_{r_0}]_{si}^{\top}[\mathbf{K}]_{ik}[\mathbf{U}_{r_0}]_{ks}\right)$. We remark that $\overline{\|\mathbf{K}\|_{r_0}}$ is ready to be optimized by standard SGD algorithms because it is separable and expressed as the summation of losses on individual training data points. We perform the optimization for low-rank transductive few-shot node classification in (2) with the approximate TNN $\overline{\|\mathbf{K}\|_{r_0}}$ instead of the original TNN $\|\mathbf{K}\|_{r_0}$.

