# OpenReview forum: "Low-Rank Few-Shot Node Classification by Node-Level Graph Diffusion"
_ICLR.cc/2026/Conference — ICLR 2026 Poster_

### Official Review · Reviewer_ocH3 · 2025-10-28

**Soundness:** 2
**Presentation:** 2
**Contribution:** 2
**Rating:** 4
**Confidence:** 4

**Summary:**

This paper focuses on few-shot node classification (FSNC) tasks with generative model-based data augmentation method. Specifically, the proposed LR-FGDM method adopts the diffusion model-based few-shot learning paradigm and serves as a plug-in module to enhance existing FSNC frameworks. The FGDM model utilizes a hierachical graph autoencoder to obatin label prototypes and synthesize graph attributes for original graph augmentation. A low-rank few-shot classification module is designed to reduce inductive noise from diffusion generation process. Experiments and ablation studies verify the effectiveness of LR-FGDM.

**Strengths:**

1. The proposed LR-FGDM is a plug-in-and-play framework that can be integrated into multiple FSNC models.
2. It is a reasonable idea to apply low-frequency filters to avoid additional noise from generative model.
3. Comprehensive experimental studies on the model's FSNC performance.

**Weaknesses:**

1. The overall contribution appears somewhat incremental, as the proposed framework mainly combines elements from prior works on diffusion on graphs (DoG) and graph augmentation–based few-shot learning.
2. Although the authors claim that the low-rank regularization ensures an upper bound on the Frobenius norm of the predicted labels, the theoretical bound presented in Eq 3 is not illustrative enough to reflect its convergence characteristics.
3. While the execution time for training and generation in FGDM is reported, the comparative training efficiency against baseline models remains unexplored.
4. The paper would benefit from a more detailed introduction to the overall FSNC algorithm and a broader discussion on the range of FSNC frameworks to which LR-FGDM can be applied.

**Questions:**

Please refer to the weaknesses.

---

> ### Author Response · Authors · 2025-11-21
> **Response to Official Review by Reviewer ocH3 (Part 1)**
>
> We appreciate the review and the suggestions in this review. The raised issues are addressed below.
>
> **Responses to the Weaknesses**
>
> **1. "The overall contribution appears somewhat incremental...prior works on diffusion on graphs (DoG) and graph augmentation–based few-shot learning."**
>
> We respectfully disagree that the contribution is incremental. The node-level graph augmentation is an active research direction with many works. DoG represents only one specific approach designed for generic node classification, rather than for Few-Shot Node Classification (FSNC). Traditional class-conditional diffusion models typically condition on class labels, including DoG, as illustrated in Figure 3 (a) in Section E of the revised paper. However, in FSNC, the classes in the support and query sets are novel and disjoint from those used during training. As the diffusion model is trained prior to test-time adaptation, it cannot directly condition on these unseen class labels.
>
> To overcome this limitation, we leverage the prototypical regularization introduced in the HGAE, which encourages the node latent representations to cluster around their respective prototype representations. HGAE jointly learns latent representations and semantically aligned prototype clusters, which enables both robust prototype-based conditioning for the latent diffusion model and an efficient hierarchical edge reconstruction mechanism.
>
> Furthermore, inspired by the generalization bound for the test loss in Theorem A.1 of the revised paper (Theorem 3.1 of the original paper),  we proposed LRA-LR-FGDM in Section 3.5 of the revised paper, which further reduces the kernel complexity of LR-FGDM in the revised paper with better performance than the original LR-FGDM. Extended experimental results in Tables 1, 2, 6, and 10 of the revised paper show that LRA-LR-FGDM even outperforms LR-FGDM. Table 9 in Section G.3 of the revised paper shows that LRA-LR-FGDM exhibits a lower KC and a lower upper bound for the test loss than that of LR-FGDM.
>
> **2. "...the theoretical bound presented in Eq 3 is not illustrative enough to reflect its convergence characteristics."**
>
>
> Equation (4) in our paper provides an upper bound for the test loss, that is, the Frobenius norm of the prediction error of the few-shot linear classifier, where the bound consists of two key quantities: the kernel complexity $\text{KC}(\mathbf{K})$ of the kernel Gram matrix and the training loss $L(\mathbf{K}, \mathbf{Y_{FS}}, t)$ with respect to the ground-truth labels. To make the convergence behaviour of this bound more explicit and empirically interpretable, we have substantially expanded the analysis in the revised paper. In Table 9 in Section G.3 of the revised paper, we report the values of bound $\text{KC}(\mathbf{K})$, $L(\mathbf{K},\mathbf{Y}_{\textup{FS}},t)$, and the resulting upper bound value from Theorem A.1, for 5-way 5-shot tasks on all the graph benchmark datasets in this paper.  **Both LR-FGDM and LRA-LR-FGDM exhibit a lower upper bound for the test loss compared to that of the two best baselines for FSNC (COSMIC and COLA), which evidence the better convergence characteristics of our methods compared to these baselines. Furthermore, LRA-LR-FGDM has an even lower upper bound for the test loss than that of LR-FGDM, showing the effectiveness of our main theoretical result, Theorem A.1, in guaranteeing the performance of FSNC.**
>
> These results directly demonstrate how the low-rank regularization influences each term during optimization. Across all datasets and for both baselines (COSMIC and COLA), their LR-FGDM-augmented and LRA-FGDM-augmented counterparts consistently achieve substantially lower kernel complexity and significantly reduced training loss, leading to a tighter upper bound on the test loss. This empirical trend directly aligns with the theoretical motivation that the low-rank regularization encourages the learned Gram matrix to have a small truncated nuclear norm, thereby reducing the Kernel Complexity $\text{KC}(\mathbf{K})$, which in turn reduces the generalization bound for the test loss in Equation (3). The observed decrease of both $\text{KC}(\mathbf{K})$ and $L(\mathbf{K},\mathbf{Y}_{\textup{FS}},t)$ across all datasets demonstrates that the theoretical bound is indeed informative and reflective of the practical convergence behaviour of LR-FGDM and LRA-FGDM.

---

> > ### Author Response · Authors · 2025-11-21
> > **Response to Official Review by Reviewer ocH3 (Part 2)**
> >
> > **3. "While the execution time for training and generation in FGDM is reported, the comparative training efficiency against baseline models remains unexplored."**
> >
> > The training time comparison between LR-FGDM and the baseline models is shown in the table below. The comparison has also been added to Table 22 in Section G.13 of the revised paper, which compares the training of baseline models with  COSMIC [12] and COLA [13] on the augmented graph generated by LR-FGDM. It is observed that LR-FGDM and LRA-LR-FGDM significantly outperform the best competing baseline, COLA, with marginal training overhead (+3.16%–3.90% for LR-FGDM and +5.17%–6.08% for LRA-LR-FGDM), with a substantial increase in accuracy (+1.83%–2.29% for LR-FGDM and +2.49%–2.85% for LRA-LR-FGDM). Table 3 of the revised paper presents the training and generation times for HGAE and LDM in LR-FGDM.
> >
> > | Method                 | CoraFull Time | CoraFull Acc | Coauthor-CS Time | Coauthor-CS Acc | ogbn-arxiv Time | ogbn-arxiv Acc | DBLP Time | DBLP Acc |
> > |---|---|---|----|---|---|---|--|---|
> > | ProtoNet               | 19.4          | 51.54%       | 15.8              | 49.25%           | 40.6             | 53.31%          | 22.9      | 52.26%    |
> > | Meta-GNN               | 83.3          | 70.50%       | 68.7              | 68.59%           | 125.1            | 31.52%          | 92.4      | 72.15%    |
> > | GPN                    | 53.0          | 72.82%       | 42.4              | 81.79%           | 171.8            | 50.50%          | 63.7      | 75.46%    |
> > | G-Meta                 | 662.5         | 75.84%       | 542.1             | 74.18%           | 598.4            | 47.16%          | 328.9     | 75.82%    |
> > | TENT                   | 58.9          | 70.10%       | 47.8              | 76.90%           | 84.2             | 61.38%          | 71.6      | 74.84%    |
> > | KD-FSNC                | 74.6          | 85.89%       | 60.8              | 84.42%           | 112.3            | 64.91%          | 83.5      | 83.75%    |
> > | NormProp               | 22.3          | 85.47%       | 18.1              | 84.61%           | 41.9             | 64.28%          | 25.8      | 84.12%    |
> > | STAR                   | 267.4         | 87.31%       | 217.6             | 87.60%           | 398.5            | 66.98%          | 302.7     | 87.10%    |
> > | DoG                    | 850.5         | 86.47%       | 690.2             | 87.35%           | 1225.3           | 65.69%          | 1042.4    | 87.59%    |
> > | COSMIC                 | 824.9         | 86.34%       | 668.2             | 85.47%           | 1221.7           | 65.42%          | 1015.3    | 85.31%    |
> > | COLA                   | 781.6         | 87.83%       | 632.3             | 87.54%           | 1155.1           | 67.52%          | 958.6     | 87.23%    |
> > | **COSMIC (LR-FGDM)**        | 854.2         | 88.81%       | 691.3             | 87.72%           | 1266.5           | 67.59%          | 1056.3    | 87.44%    |
> > | **COSMIC (LRA-LR-FGDM)**    | 875.3         | 89.47%       | 704.4             | 88.21% | 1295.1           | 68.11% | 1078.1    | 87.96%    |
> > | **COLA (LR-FGDM)**          | 810.5         | 89.66%       | 652.3             | 89.83%           | 1200.2           | 69.63%          | 992.4     | 89.51%    |
> > | **COLA (LRA-LR-FGDM)**      | 829.2         | 90.32%       | 670.4             | 90.39%  | 1221.5           | 70.22% | 1008.2    | 90.07%    |
> >
> >
> >
> >
> > **4. "The paper would benefit from a more detailed introduction to the overall FSNC algorithm and a broader discussion on the range of FSNC frameworks to which LR-FGDM can be applied."**
> >
> > A systematic pipeline of the standard FSNC workflow typically consists of (1) computing node representations from a pretrained or self-supervised graph encoder, (2) constructing a small support set for each class, (3) training a transductive linear classifier on the support set while leveraging unlabeled nodes, and (4) performing query prediction using the learned classifier. LR-FGDM is positioned as an augmentation module that acts after representation extraction and before final label prediction.
> >
> > We remark that LR-FGDM can be applied to all the FSNC frameworks, including those discussed in Section 3.1. This is because LR-FGDM operates directly on the kernel Gram matrix of node representations and optimizes the test-loss upper bound of the transductive linear classifier, so it is compatible with any FSNC method that outputs node embeddings or class logits. This includes prototype-based and metric-based methods such as ProtoNet [1], Meta-GNN [2], GPN [3], G-Meta [4], TENT [5], KD-FSNC [6], NormProp [7], and STAR [8]; contrastive or clustering-based methods such as CPLAE [9], Meta-Contrastive [10], and PsCo [11]; transductive metric-based approaches such as COSMIC [12]; and contrastive prototype-based methods such as COLA [13].

---

> > > ### Author Response · Authors · 2025-11-21
> > > **References in the Response to Official Review by Reviewer ocH3**
> > >
> > > **References**
> > >
> > > [1] Snell, Jake, et al. “Prototypical networks for few-shot learning.” NeurIPS. 2017.
> > >
> > > [2] Zhou, Fan, et al. “Meta-GNN: On few-shot node classification in graph meta-learning.” CIKM. 2019.
> > >
> > > [3] Ding, Kaize, et al. “Graph prototypical networks for few-shot learning on attributed networks.” CIKM. 2020.
> > >
> > > [4] Huang, Wenxuan, et al. “G-Meta: Graph meta-learning via local subgraphs.” NeurIPS. 2020.
> > >
> > > [5] Wang, Song, et al. “Task-adaptive few-shot node classification.” KDD. 2022.
> > >
> > > [6] Wu, Zongqian, et al. “Self-training based few-shot node classification by knowledge distillation.” AAAI. 2024.
> > >
> > > [7] Zhang, Baoming, et al. “Normalize then propagate: Efficient homophilous regularization for few-shot semi-supervised node classification.” AAAI. 2025.
> > >
> > > [8] Liu, Yonghao, et al. “Enhancing unsupervised graph few-shot learning via set functions and optimal transport.” KDD. 2025.
> > >
> > > [9] Gao, Yizhao, et al. “Contrastive prototype learning with augmented embeddings for few-shot learning.” UAI. 2021.
> > >
> > > [10] Ni, Renkun, et al. “The close relationship between contrastive learning and meta-learning.” ICLR. 2021.
> > >
> > > [11] Jang, Eunseong, et al. “PsCo: Prototype-based supervised contrastive learning for few-shot meta-learning.” ICLR. 2023.
> > >
> > > [12] Wang, Xiyuan, et al. “Contrastive meta-learning for few-shot node classification.” KDD. 2023.
> > >
> > > [13] Liu, Xuhui, et al. “Graph contrastive learning meets graph meta learning: A unified method for few-shot node tasks.” WWW. 2024.

---

### Official Review · Reviewer_Kffe · 2025-10-29

**Soundness:** 4
**Presentation:** 4
**Contribution:** 3
**Rating:** 8
**Confidence:** 4

**Summary:**

This isn't a paper in my field, but it appears to be a very solid piece of work. It includes derivations, well-conducted experiments, comparisons with recent baselines, and open-sourced code, a good paper.

The paper proposes LR-FGDM, a novel and well-motivated framework for few-shot node classification that combines node-level graph diffusion with low-rank transductive classification. Its key strengths include: (1) a prototype-conditioned graph diffusion model (FGDM) that generates both synthetic nodes and realistic edges without relying on unavailable class labels during test time; (2) an efficient hierarchical edge reconstruction mechanism that reduces decoding complexity; (3) a theoretically grounded low-rank regularization based on the Low Frequency Property, supported by a generalization bound; and (4) comprehensive experiments across diverse datasets—including heterophilic graphs—showing consistent improvements over state-of-the-art methods, along with new evaluation metrics (FND/FED) to assess generation fidelity.

**Strengths:**

The paper proposes LR-FGDM, a novel and well-motivated framework for few-shot node classification that combines node-level graph diffusion with low-rank transductive classification. Its key strengths include: (1) a prototype-conditioned graph diffusion model (FGDM) that generates both synthetic nodes and realistic edges without relying on unavailable class labels during test time; (2) an efficient hierarchical edge reconstruction mechanism that reduces decoding complexity; (3) a theoretically grounded low-rank regularization based on the Low Frequency Property, supported by a generalization bound; and (4) comprehensive experiments across diverse datasets—including heterophilic graphs—showing consistent improvements over state-of-the-art methods, along with new evaluation metrics (FND/FED) to assess generation fidelity.

**Weaknesses:**

Despite its strengths, the work has several limitations. First, the comparison with DoG is potentially unfair, as DoG is adapted using pseudo-labels that may not reflect its optimal use in few-shot settings. Second, the computational cost of the truncated nuclear norm (TNN) regularization—particularly the need for SVD—is not adequately addressed, raising scalability concerns. Third, while FND/FED quantify distributional similarity, the semantic correctness of generated nodes lacks intuitive validation (e.g., via embedding visualization). Fourth, the reason for strong performance on heterophilic graphs like Roman-Empire is not deeply analyzed. Finally, certain implementation details (e.g., SVD approximation, prototype initialization) remain unclear, which could affect reproducibility.

**Questions:**

see above.

---

> ### Author Response · Authors · 2025-11-21
> **Response to Official Review by Reviewer Kffe (Part 1)**
>
> We appreciate the review and the suggestions in this review. The raised issues are addressed below.
>
> **Responses to the Weaknesses**
>
> **1. "First, the comparison with DoG is potentially unfair, as DoG is adapted using pseudo-labels that may not reflect its optimal use in few-shot settings."**
>
> To better verify the advantages of LR-FGDM over DoG, we have strengthened the DoG by constructing a new variant, DoG-prototype, specifically tailored for Few-Shot Node Classification (FSNC). Unlike the original DoG implementation, which relies on pseudo-labels obtained from semi-supervised K-means, as used in our LR-FGDM, DoG-prototype uses the learned prototypes with the GAE as conditioning signals during diffusion training and generation. This design provides DoG with a conditioning mechanism that is significantly more stable than pseudo labels and better aligned with the FSNC regime, which offers a much fairer point of comparison. The study is conducted on the most challenging dataset,ogbn-arxiv, using COLA as the base FSNC method.
> Results in the table below show that LR-FGDM still outperforms DoG-prototype by $1.18%$, which demonstrates that the improvement cannot be attributed to the prototype-based conditioning alone. Instead, the gains of LR-FGDM stem from the joint benefit of our hierarchical edge reconstruction design, the prototypical regularization for better semantic alignment, and the theoretically motivated low-rank learning that reduces kernel complexity for improved generalization capability.
>
> |    Methods    | Accuracy (ogbn-arxiv) |
> | :----: | :--: |
> | DoG-prototype |        68.45%         |
> |    LR-FGDM    |        69.63%         |
>
> **2. "Second, the computational cost of the truncated nuclear norm (TNN) regularization—particularly the need for SVD—is not adequately addressed, raising scalability concerns."**
>
> Response: We introduced the Nystrom method to improve the efficiency of computing the TNN in our experiments. The details of the  Nystrom method are described in Section I of the appendix of the revised paper.
> The efficient Nystrom method estimates the top-$r_0$ eigenvectors of the Gram matrix $\mathbf{B}$ without performing a full SVD. Instead, we sample a small set of landmark points to construct a low-dimensional Gram submatrix and compute its top-$r_0$ eigendecomposition. The resulting eigenvectors are then extended to the full dataset via the Nystrom extension, leading to an accurate approximation of the dominant eigenspace with substantially reduced time and memory complexity. Such an efficient Nystrom method for the computation of the TNN has been used for all the experiments in this paper.
>
>
> **3. "Third, while FND/FED quantify distributional similarity, the semantic correctness of generated nodes lacks intuitive validation (e.g., via embedding visualization)."**
>
> Response: To provide a more intuitive assessment of the semantic correctness of the generated nodes, we have added t-SNE visualizations in Figure 7 of the revised paper. The visualization results illustrate the spatial distribution of the node embeddings of the synthetic nodes generated by LR-FGDM compared to the node embeddings of the real nodes in the corresponding few-shot classes. As shown in Figure 7 of the revised paper, embeddings of the synthetic nodes produced by LR-FGDM form compact clusters aligned closely with the real node clusters, demonstrating that the generated nodes preserve both semantic structure and class-level discriminativeness. These qualitative results complement the quantitative FND/FED metrics and provide additional evidence that LR-FGDM generates semantically meaningful node representations.
>
> **4. "Fourth, the reason for strong performance on heterophilic graphs like Roman-Empire is not deeply analyzed."**
>
> The reason for the strong performance of our LR-FGDM is due to the reduced Kernel Complexity (KC) by low-rank regularization, and we show in Table 9 of the revised paper that LR-FGDM exhibits lower KC and lower upper bound for the test loss than that of the competing baselines on all graph datasets, including both the homophilic and the heterophilic graph datasets, including Roman-Empire. Furthermore, the LRA-LR-FGDM proposed in the revised paper even improves the performance of LR-FGDM by further reducing the KC of LR-FGDM, with the performance of LRA-LR-FGDM in Tables 1, 2, 6, and 10 of the revised paper. Table 9 of the revised paper shows that LRA-LR-FGDM exhibits even lower KC and a lower upper bound for the test loss than LR-FGDM and the competing baselines.

---

> > ### Author Response · Authors · 2025-11-21
> > **Response to Official Review by Reviewer Kffe (Part 2)**
> >
> > **5. "Finally, certain implementation details (e.g., SVD approximation, prototype initialization) remain unclear, which could affect reproducibility."**
> >
> > As detailed in our response to weakness 2, we introduced the efficient Nystrom method for the computation of the TNN in Section I of the appendix of the revised paper. Details about the implementation and the hyperparameters of the Nystrom method are also added to Section I of the appendix of the revised paper.
> >
> > The first epoch in the training of the HGAE is a warm-up process where the prototypical loss is not optimized in the first epoch. After the warm-up process, prototypes are computed as K-means cluster centroids based on the latent embeddings produced by the HGAE encoder over all nodes in the graph, and the prototypical loss is then optimized in all the subsequent epochs. Such a warm-up process ensures that meaningful prototypes obtained after the first epoch are used in the prototypical loss.
> >
> >
> > **References**
> >
> > [1] Sanjiv Kumar, Mehryar Mohri, and Ameet Talwalkar. Sampling methods for the nystrom method. J. Mach. Learn. Res., 13:981–1006, 2012

---

### Official Review · Reviewer_678A · 2025-10-29

**Soundness:** 4
**Presentation:** 3
**Contribution:** 3
**Rating:** 6
**Confidence:** 3

**Summary:**

This paper proposes a framework for few-shot node classification that uses a node-level graph diffusion model to generate synthetic support nodes and edges for augmenting the support set, and introduces low-rank regularization to mitigate diffusion-induced noise, with theoretical guarantees for transductive classification; extensive experiments on 8 datasets show LR-FGDM outperforms sota FSNC methods.

**Strengths:**

- LR-FGDM solves FSNC data scarcity by generating synthetic support nodes and edges.
- The low-rank regularization in LR-FGDM fixes diffusion-induced noise, makeing the quality of synthetic data more reliable for model training.
- The paper provides theoretical proof for the low-rank regularization, supporting the method’s validity with mathematical generalization bounds.
- The proposed model outperforms state-of-the-art FSNC methods consistently, showing real effectiveness across 8 benchmark datasets.

**Weaknesses:**

- For the hierarchical edge reconstruction, \(d_i\) (number of connected clusters per node) is bounded by node degree, but what if nodes have extremely high degrees (e.g., hub nodes in scale-free graphs)?
- The low-rank regularization relies on truncated nuclear norm (\(\|K\|_{r_0}\)), but \(r_0\) is selected via cross-validation—why not use adaptive rank selection (e.g., based on eigenvalue decay rate of K) to avoid manual tuning?
- When generating synthetic edges, the HGAE decodes intra-cluster neighbor maps, but how to ensure synthetic edges follow real graph structural properties (e.g., small-world, power-law)?
- The LDM is conditioned on prototypes instead of labels, but prototypes are learned from base classes, does prototype shift (between base and novel classes) reduce synthetic node quality for novel classes, and how to mitigate this?

**Questions:**

See Weaknesses

---

> ### Author Response · Authors · 2025-11-21
> **Response to Official Review by Reviewer 678A (Part 1)**
>
> We appreciate the review and the suggestions in this review. The raised issues are addressed below.
>
> **Responses to the Weaknesses**
>
> **1. "... what if nodes have extremely high degrees (e.g., hub nodes in scale-free graphs)?"**
>
> Our HGAE does not impose any constraint on the node degree. The design of HGAE explicitly supports nodes with arbitrarily large degrees, including hub nodes commonly found in scale-free graphs. This is because edge decoding is performed in a hierarchical manner with no constraint on $d_i$, and each node can be connected to all the node clusters. The edge decoder first predicts connected prototype-induced node clusters, then reconstructs the corresponding intra-cluster neighbor map within every connected node cluster. As a result, every node can connect to all the clusters, and our HGAE can handle nodes of arbitrarily high degrees.
>
> **2. "The low-rank regularization relies on truncated nuclear norm $||\mathbf{K}||_{r_0}$, but $r_0$ is selected via cross-validation—why not use adaptive rank selection..."**
>
> We explore adaptive rank-selection strategies based on the eigenvalue decay of the feature kernel Gram matrix $\mathbf{K}$.
> We also evaluate an eigengap-based method widely used in spectral dimensionality selection, in which $r_0$ is selected at the index corresponding to the largest relative decay between consecutive eigenvalues [2]. In particular, $r_0$ is computed as $r_0 = \arg \max_{1 \le i < N}$ $\frac{\hat\lambda_i - \hat{\lambda}_{i+1}}{\hat{\lambda}_i}$.
> The results are shown in the table below. The performance of LR-FGDM and LRA-LR-FGDM with the automatically selected rank $r_0$ is only marginally different from that achieved via cross-validation. The deviations remain consistently marginal, typically smaller than $0.2\%$. The results suggest that the automatic rank-selection strategy based on the eigenvalue decay rate provides an efficient alternative to tuning the rank $r_0$, without sacrificing performance. The study has been added to Section G.10 in the appendix of the revised paper.
>
> | Method   | Selection Method of $r_0$ | CoraFull | ogbn-arxiv | Coauthor-CS | DBLP  |
> |------|----|----------|---|------|---|
> | COLA                    | –                             | 87.83    | 67.52      | 87.54        | 87.23 |
> | COLA (LR-FGDM)              | Cross-Validation              | 89.66 | 69.63  | 89.83    | 89.51 |
> | COLA (LR-FGDM)              | Automatic                     | 89.56    | 69.68      | 89.74        | 89.55 |
> | COLA (LRA-LR-FGDM)          | Cross-Validation              | 90.32 | 70.22  | 90.39   | 90.07 |
> | COLA (LRA-LR-FGDM)          | Automatic                     | 90.36    | 70.04      | 90.21        | 90.11 |
>
>
>
> **3. "When generating synthetic edges, the HGAE decodes intra-cluster neighbor maps, but how to ensure synthetic edges follow real graph structural properties (e.g., small-world, power-law)?"**
>
> As discussed in our response to weakness 1, the HGAE is trained to reconstruct both node attributes and edges by minimizing the reconstruction loss, which forces the reconstructed edges to match the structural patterns observed in the real graph. To further ensure that the synthetic edges preserve key graph properties, we evaluate two quantitative graph structural properties, including
> (1) small-world property [3], which is measured by the small-world coefficient  $\sigma = \frac{C/C_{\text{rand}}}{L/L_{\text{rand}}}$ where $C$ and $L$ denote the clustering coefficient and characteristic path length of the augmented graph, and $C_{\text{rand}}, L_{\text{rand}}$ are the clustering coefficient and characteristic path length a degree-preserving random graph.
> (2) power-law degree distribution [4, 5], which is measured by the power-law exponent and the Kolmogorov–Smirnov (KS) statistic between the empirical degree distribution and a fitted power-law.
> As shown in the table below, the synthetic graph structures generated by LR-FGDM exhibit a small-world coefficient $\sigma$ and power-law exponent $\alpha$ that closely match those of the original graph, with only marginal deviation in the KS statistic. These results confirm that the hierarchical edge reconstruction in HGAE successfully preserves both the small-world characteristics and the heavy-tailed degree patterns of real-world graphs. The above study has also been added to Section G.14 of the appendix of the revised paper.
>
> | Metric | Graph     | CoraFull | ogbn-arxiv | Coauthor-CS | DBLP  |
> |--------|-----------|----------|------------|--------------|-------|
> | $\sigma$ | Original  | 3.27     | 2.56       | 4.11         | 2.99  |
> | $\sigma$ | Augmented | 3.12     | 2.48       | 3.99         | 2.94  |
> | KS     | Original  | 0.067    | 0.054      | 0.062        | 0.049 |
> | KS     | Augmented | 0.068    | 0.056      | 0.061        | 0.052 |

---

> > ### Author Response · Authors · 2025-11-21
> > **Response to Official Review by Reviewer 678A (Part 2)**
> >
> > **4. "The LDM is conditioned on prototypes instead of labels, but prototypes are learned from base classes...?"**
> >
> > In Few-Shot Node Classification (FSNC), the prototypes are learned from the features of nodes from both the base classes and the novel classes in an unsupervised manner, following the literature in FSNC [6, 7, 8, 9]. As a result, LR-FGDM does not rely on fixed base-class prototypes and therefore does not suffer from prototype shift between base and novel classes.
> >
> > **References**
> >
> > [1] Schölkopf, Bernhard, Alexander Smola, and Klaus-Robert Müller. "Nonlinear component analysis as a kernel eigenvalue problem." Neural computation 10.5 (1998): 1299-1319.
> >
> > [2] Ng, Andrew, Michael Jordan, and Yair Weiss. "On spectral clustering: Analysis and an algorithm." Advances in neural information processing systems 14 (2001).
> >
> > [3] Telesford, Qawi K., et al. "The ubiquity of small-world networks." Brain connectivity 1.5 (2011): 367-375.
> >
> > [4] Clauset, Aaron, Cosma Rohilla Shalizi, and Mark EJ Newman. "Power-law distributions in empirical data." SIAM review 51.4 (2009): 661-703.
> >
> > [5] ​​Zhao, Tianyi, Jian Kang, and Lu Cheng. "Conformalized link prediction on graph neural networks." Proceedings of the 30th ACM SIGKDD Conference on Knowledge Discovery and Data Mining. 2024.
> >
> > [6] Wu, Zongqian, et al. "A pseudo-labeling approach based on knowledge distillation for graph few-shot learning." Information Processing & Management 62.6 (2025): 104268.
> >
> > [7] Wu, Zongqian, et al. "Self-training based few-shot node classification by knowledge distillation." Proceedings of the AAAI conference on artificial intelligence. Vol. 38. No. 14. 2024.
> >
> > [8] Ding, Kaize, et al. "Meta propagation networks for graph few-shot semi-supervised learning." Proceedings of the AAAI conference on artificial intelligence. Vol. 36. No. 6. 2022.
> >
> > [9] Wu, Zongqian, et al. "Information Augmentation for Few-shot Node Classification." IJCAI. 2022.

---

### Official Review · Reviewer_dzx3 · 2025-10-30

**Soundness:** 3
**Presentation:** 2
**Contribution:** 2
**Rating:** 4
**Confidence:** 3

**Summary:**

This paper proposes Low-Rank Few-Shot Graph Diffusion Model (LR-FGDM) for few-shot node classification (FSNC). LR-FGDM augments the limited support set by generating realistic synthetic nodes and edges through a node-level graph diffusion generator (FGDM), which combines a Hierarchical Graph Autoencoder (HGAE) with prototypical regularization and a prototype-conditioned Latent Diffusion Model (LDM). To mitigate noise from diffusion generation, a low-rank regularization inspired by the Low-Frequency Property (LFP) is applied to the transductive classifier, with theoretical guarantees on reduced kernel complexity and generalization error. Extensive experiments across multiple graph benchmarks demonstrate that LR-FGDM consistently outperforms state-of-the-art FSNC methods in both accuracy and structural fidelity.

**Strengths:**

- The paper introduces a novel and well-motivated framework for few-shot node classification that combines prototype-conditioned diffusion-based augmentation with low-rank regularization.
- The idea of performing node-level diffusion generation, as opposed to graph-level synthesis, is original and provides a meaningful contribution to generative graph learning.
- The theoretical analysis grounding the low-rank transductive classifier adds rigor and clarity, while the empirical results demonstrate consistent improvements over strong baselines across multiple benchmarks.

**Weaknesses:**

- The pipeline figure could be substantially improved. It is visually cluttered and does not clearly separate the HGAE, diffusion, and low-rank classification stages, making it difficult to follow the data flow.

- The length allocation of the paper is also unbalanced: the main text devotes extensive space to model formulation while leaving only two pages for experiments, which contain no figures and limited analysis. Important aspects such as efficiency comparisons and runtime scalability should be included in the main text rather than deferred to the appendix.

- Table 2 suggests that the performance gain from the model’s architectural design may be relatively small, and Table 6 shows that the p-values for COSMIC (LR-FGDM) remain relatively high, raising concerns about the statistical robustness of the reported improvements.

- The paper lacks an ablation study for the diffusion (LDM) component, making it difficult to assess how much of the improvement originates from the proposed generative mechanism rather than from low-rank or prototypical regularization.

**Questions:**

1. On the Theoretical Scope of Theorem 3.1.
Could the authors explicitly state the assumptions underlying Theorem 3.1 and clarify under what conditions the generalization bound holds? Specifically, does the theorem rely on boundedness, Lipschitz continuity, or i.i.d. sampling assumptions, and is it theoretically valid for all graph structures or only for homophilic/community-structured graphs?

2. On the Missing Diffusion Ablation and Efficiency Evidence.
While Table 2 ablates the low-rank and prototypical regularizations, the LDM module itself is never directly examined. Could the authors include a variant without the diffusion process (e.g., using only the VAE) or with a generic diffusion without the VAE, and report runtime and memory comparisons with COSMIC and DoG to clarify whether the generative component justifies its computational overhead?

---

> ### Author Response · Authors · 2025-11-21
> **Response to Official Review by Reviewer dzx3 (Part 1)**
>
> We appreciate the review and the suggestions in this review. The raised issues are addressed below.
>
> **Responses to the Weaknesses**
>
> **1. "The pipeline figure could be substantially improved...making it difficult to follow the data flow."**
>
> Figure 2 (b) is intended solely to illustrate the HGAE with prototypical regularization and the prototype-conditioned LDM, instead of the complete LR-FGDM workflow. In the revised paper, we have added a new Figure 2 (a) illustrating the full pipeline, including the training of the HGAE with prototypical regularization, the training of the LDM, the generation of the synthetic graph structure (SGS), and the training of the low-rank transductive classifier with the low-rank regularization.
> The original Figure 2 (a), which compares LR-FGDM and DoG, has been moved to Figure 3 in Section E of the appendix.
>
> **2. "The length allocation of the paper is also unbalanced...Important aspects such as efficiency comparisons and runtime scalability should be included in the main text rather than deferred to the appendix."**
>
> We have moved the detailed theoretical justification for the low-rank regularization to Section A of the appendix in the revised paper. In addition, the efficiency analysis of LR-FGDM for both training and SGS generation has been moved to Section 4.4 of the revised main paper.
>
> **3. "Table 2 suggests that the performance gain ... robustness of the reported improvements."**
>
> The primary novel architectural design in LR-FGDM is the HGAE with prototypical regularization. We emphasize that both low-rank regularization and prototypical regularization, along with the architectural design of the HGAE in our LR-FGDM, are all important contributions of this paper. We do not aim to maximize the benefit of the architectural design of HGAE alone. Instead, we seek the maximal improvement combining the architectural design of the HGAE, prototypical regularization, and low-rank regularization.
> The important contribution of the HGAE lies in its improvement of generation efficiency, which is achieved by a prototype-guided hierarchical edge reconstruction mechanism that enables efficient and semantically coherent generation of synthetic graph structures, rather than explicitly maximizing Few-Shot Node Classification (FSNC) performance on its own. The efficiency of HGAE has been analyzed in Section C of the appendix and empirically validated in Table 3 in Section 4.4 of the revised paper.
>
> The models enhanced by LR-FGDM consistently exhibit statistically significant improvements over the corresponding baselines across all few-shot settings, as indicated by the p-values below 0.05 reported in Table 6 in Section G.1 of the appendix. The use of the 0.05 threshold follows the widely accepted textbook-level standard for assessing statistical significance with the Student’s t-test [1].
>
> Furthermore, inspired by the generalization bound for the test loss in Theorem A.1 of the revised paper (Theorem 3.1 of the original paper),  we have proposed LRA-LR-FGDM in Section 3.5 of the revised paper, which further reduces the kernel complexity of LR-FGDM in the revised paper with better performance than the original LR-FGDM. Extended experimental results in Table 1, Table 2, Table 6, and Table 10  of the revised paper show that LRA-LR-FGDM even outperforms LR-FGDM, and Table 7, Table 11, and Table 17 of the revised paper show that LRA-LR-FGDM exhibits even lower t-test p-values against the baseline models than those of LR-FGDM. The largest p-value across all graph datasets of LR-FGDM is $0.044< 0.05$, and that of LRA-LR-FGDM is $0.010< 0.05$,  which indicates statistically significant improvements.
>
> **4. "The paper lacks an ablation study for the diffusion (LDM) component, making it difficult to assess how much of the improvement originates from the proposed generative mechanism rather than from low-rank or prototypical regularization."**
>
> To demonstrate the contribution of the proposed generative mechanism, we have performed an additional ablation study in which the LDM is replaced with two alternative generative mechanisms, including a Variational Autoencoder (VAE) [2] and a generic diffusion model, DDPM [3], with results shown below and also in Table 21 of the revised paper.
>
> | Method    | CoraFull | ogbn-arxiv | Coauthor-CS | DBLP  |
> |--|------|----|-------|-
> | COLA | 87.83    | 67.52      | 87.54       | 87.23 |
> | COLA (DDPM)         | 88.85    | 68.35      | 88.42       | 88.12 |
> | COLA (VAE)          | 88.47    | 68.26      | 88.30       | 88.07 |
> | **COLA (LR-FGDM)**  | **89.56** | **69.68**  | **89.74**   | **89.55** |

---

> > ### Author Response · Authors · 2025-11-21
> > **Response to Official Review by Reviewer dzx3 (Part 2)**
> >
> > **Responses to the Questions**
> >
> > **1. On the Theoretical Scope of Theorem 3.1. Could the authors explicitly state the assumptions ... is it theoretically valid for all graph structures or only for homophilic/community-structured graphs?**
> >
> > Theorem 3.1 in the original paper (Theorem A.1 of the revised paper), referred to as the main theorem, only assumes that the labeled training set of size $m$ is randomly sampled uniformly from all the $N$ nodes of the graph without replacement, the MSE loss is bounded, and $m \ge cN$ for a constant $c$ where. Such assumptions are very mild and standard in the FSNC and graph learning/transductive learning literature. In particular, uniform sampling of the labeled training set without replacement is widely adopted in FSNC [4, 5, 6] and semi-supervised/transductive node classification in [7]. The assumptions about bounded MSE loss and  $m \ge cN$ are used in the generalization analysis of standard transductive learning in [8,9].
> >
> > We remark that our assumptions do not require homophily or community-structured conditions. As a result, the generalization bound in our main theorem is applicable to all graph structures, including both homophilic/community-structured and heterophlic graphs. Such a claim is evidenced empirically by our experiments.
> > We have also demonstrated the effectiveness of LR-FGDM on both homophilic graph datasets, as reported in Table 1 and Table 6, and heterophilic graph datasets, as shown in Table 10 in Section G.4 of the revised paper. Across all settings, FSNC methods equipped with the augmented graphs generated by LR-FGDM substantially improve the performance of the corresponding baseline models. Furthermore, motivated by our main theorem, we introduce LRA-LR-FGDM in the revised paper, which further strengthens the performance of LR-FGDM by reducing kernel complexity and tightening the generalization error bound for the test loss. These improvements are consistently supported by the experimental results presented in Tables 1, 2, 6, and 10 of the revised paper.
> >
> >
> > **2. On the Missing Diffusion Ablation and Efficiency Evidence. While Table 2 ablates the low-rank and prototypical regularizations, the LDM module itself is never directly examined...?**
> >
> > Please refer to our response to Weakness 4.
> >
> > **References**
> >
> > [1] Casella, George, and Roger Berger. Statistical inference. Chapman and Hall/CRC, 2024.
> >
> > [2] Kingma, Diederik P., and Max Welling. "Auto-encoding variational bayes." ICLR 2014
> >
> > [3] Ho, Jonathan, Ajay Jain, and Pieter Abbeel. "Denoising diffusion probabilistic models." NeurIPS 2020.
> >
> > [4] Wu, Zongqian, et al. "Self-training based few-shot node classification by knowledge distillation." Proceedings of the AAAI conference on artificial intelligence. Vol. 38. No. 14. 2024.
> >
> > [5] Ding, Kaize, et al. "Meta propagation networks for graph few-shot semi-supervised learning." Proceedings of the AAAI conference on artificial intelligence. Vol. 36. No. 6. 2022.
> >
> > [6] Wu, Zongqian, et al. "Information Augmentation for Few-shot Node Classification." IJCAI. 2022.
> >
> > [7] Kipf, Thomas N., and Max Welling. "Semi-Supervised Classification with Graph Convolutional Network." ICLR 2017
> >
> > [8] Y. Yang. A New Concentration Inequality for Sampling Without Replacement and Its Application for Transductive Learning. ICML 2025.
> >
> > [9] Tolstikhin et al. Localized Complexities for Transductive Learning. COLT 2014.

---

### Author Response · Authors · 2025-12-02
**Summary of Revision (Part 1)**

Dear AC,

Thank you for your time handling and reviewing this paper. We have thoroughly addressed all reviewers’ concerns, which is also reflected in the revised paper. Below is a concise summary of our key clarifications and revisions addressing all the concerns.

**1. Summary of Revisions Addressing the Comments by Reviewer ocH3**

- **Reviewer ocH3 is concerned about the contribution of our LR-FGDM over prior work diffusion on graphs (DoG) [1], and that the contribution of our LR-FGDM appears incremental.**
We respectfully disagree that the contribution is incremental. The node-level graph augmentation is an active research direction with many works. DoG represents only one specific approach designed for generic node classification, rather than for Few-Shot Node Classification (FSNC). Traditional class-conditional diffusion models typically condition on class labels, including DoG, as illustrated in Figure 3 (a) in Section E of the revised paper. However, in FSNC, the classes in the support and query sets are novel and disjoint from those used during training. As the diffusion model is trained prior to test-time adaptation, it cannot directly condition on these unseen class labels.
To overcome this limitation, we leverage the prototypical regularization introduced in the HGAE, which encourages the node latent representations to cluster around their respective prototype representations. HGAE jointly learns latent representations and semantically aligned prototype clusters, which enables both robust prototype-based conditioning for the latent diffusion model and an efficient hierarchical edge reconstruction mechanism.
**As another significant contribution over DoG, We have improved the theoretical contribution of our work by a new method, LRA-LR-FGDM, in the revised paper, which further enhances the LR-FGDM in our original paper**. Inspired by the generalization bound for the test loss in Theorem A.1 of the revised paper (Theorem 3.1 of the original paper),  we have proposed LRA-LR-FGDM in Section 3.5 of the revised paper, which further reduces the kernel complexity of LR-FGDM in the revised paper with better performance than the original LR-FGDM. Extended experimental results in Table 1, Table 2, Table 6, and Table 10  of the revised paper show that LRA-LR-FGDM even outperforms LR-FGDM, and Table 7, Table 11, and Table 17 of the revised paper show that LRA-LR-FGDM exhibits even lower t-test p-values against the baseline models than those of LR-FGDM. The largest p-value across all graph datasets of LR-FGDM is $0.044< 0.05$, and that of LRA-LR-FGDM is $0.010< 0.05$,  which indicates statistically significant improvements.

- **Reviewer ocH3 is concerned that the upper bound for the test loss presented in Eq. 4 is not illustrative enough to reflect its convergence characteristics.**
To address this concern, we first clarify that Equation (4) in our paper provides an upper bound for the test loss, that is, the Frobenius norm of the prediction error of the few-shot linear classifier, where the bound consists of two key quantities: the kernel complexity $\text{KC}(\mathbf{K})$ of the kernel Gram matrix and the training loss $L(\mathbf{K},\mathbf{Y} _ {\textup{FS}},t)$ with respect to the ground-truth labels. In Table 9 in Section G.3 of the revised paper, we report the values of bound $\text{KC}(\mathbf{K})$, $L(\mathbf{K},\mathbf{Y} _ {\textup{FS}},t)$, and the resulting upper bound value from Theorem A.1.  **Both LR-FGDM and LRA-LR-FGDM exhibit a lower upper bound for the test loss compared to that of the two best baselines for FSNC (COSMIC and COLA), which evidence the better convergence characteristics of our methods compared to these baselines. Furthermore, LRA-LR-FGDM has an even lower upper bound for the test loss than that of LR-FGDM, showing the effectiveness of our main theoretical result, Theorem A.1, in guaranteeing the performance of FSNC.** For both baselines (COSMIC and COLA) and across all datasets, their LR-FGDM and LRA-FGDM augmented counterparts achieve significantly lower kernel complexity and training loss, leading to a tighter upper bound on the test loss. The consistent reductions in $\text{KC}(\mathbf{K})$ and $L(\mathbf{K},\mathbf{Y}_{\textup{FS}},t)$ further confirm that the theoretical bound is informative and accurately reflects the practical convergence behaviour of LR-FGDM and LRA-FGDM.

- **Reviewer ocH3 is concerned that the comparative training efficiency against baseline models remains unexplored.**
To address this concern, we have added the training time comparison between LR-FGDM, LRA-FGDM, and the baseline models to Table 22 in Section G.13 of the revised paper. It is observed that LR-FGDM and LRA-LR-FGDM significantly outperform the best competing baseline, COLA, with marginal training overhead (+3.16%–3.90% for LR-FGDM and +5.17%–6.08% for LRA-LR-FGDM), while achieving a substantial increase in accuracy (+1.83%–2.29% for LR-FGDM and +2.49%–2.85% for LRA-LR-FGDM).

---

> ### Author Response · Authors · 2025-12-02
> **Summary of Revision (Part 2)**
>
> - **Reviewer ocH3 is concerned that our paper does not have a detailed introduction to the overall FSNC algorithm and a broader discussion on the range of FSNC frameworks to which LR-FGDM can be applied.**
> To address this concern, we have clarified the standard FSNC pipeline, which consists of extracting node representations, forming a support set, training a transductive classifier with both labeled and unlabeled nodes, and performing query prediction. LR-FGDM functions as an augmentation module inserted after representation extraction and before label prediction. We have further clarified that LR-FGDM is compatible with all FSNC frameworks discussed in Section 3.1, since it operates directly on the kernel Gram matrix of node representations and optimizes the test-loss upper bound of the transductive classifier. As a result, it can be applied to any FSNC method that outputs node embeddings or class logits, including prototype- and metric-based methods, contrastive or clustering-based approaches, and transductive or contrastive prototype-based variants.
>
> **2. Summary of Revisions Addressing the Comments by Reviewer dzx3**
>
> - **Reviewer dzx3 is concerned that the pipeline figure is difficult to follow.**
> To address this concern, we have added a new Figure 2(a) that illustrates the full pipeline, including HGAE training with prototypical regularization, LDM training, synthetic graph structure generation, and the training of the low-rank transductive classifier with low-rank regularization.
>
> - **Reviewer dzx3 is concerned that important aspects, efficiency comparisons, and runtime scalability should be included in the main text.**
> To address this concern, we have moved the efficiency analysis of LR-FGDM for both training and SGS generation to Section 4.4 of the revised main paper and moved the detailed theoretical justification of the low-rank regularization to Section A of the appendix.
>
> - **Reviewer dzx3 is concerned that the performance gain from the model’s architectural design is relatively small, and the p-values for COSMIC (LR-FGDM) remain relatively high.**
> To address this concern, we have clarified that the key architectural novelty of LR-FGDM is the HGAE with prototypical regularization. Our contributions lie in the joint betters of the HGAE design, prototypical regularization, and low-rank regularization. The main contribution of HGAE is to improve the generation efficiency via prototype-guided hierarchical edge reconstruction, which is analyzed in Section C of the appendix and empirically validated in Table 3 in Section 4.4 of the revised paper.
> In addition, models enhanced by LR-FGDM consistently achieve statistically significant gains over their baselines across all few-shot settings, with p-values below 0.05 as reported in Table 6 (Section G.1). The 0.05 threshold follows the standard criterion for significance in Student’s t-tests [2]. Furthermore, we have proposed LRA-LR-FGDM in Section 3.5 of the revised paper with better performance than the original LR-FGDM. As discussed in point (1) of our response, LRA-LR-FGDM exhibits even lower t-test p-values against the baseline models than those of LR-FGDM.
>
> - **Reviewer dzx3 is concerned that our paper lacks an ablation study for the LDM.**
> To address this concern, we have performed an additional ablation study in which the LDM is replaced with two alternative generative mechanisms, including a VAE and a generic diffusion model, DDPM, with results shown in Table 21 of the revised paper.
>
> - **Reviewer dzx3 is concerned about the assumptions underlying Theorem 3.1 and requires clarification under what conditions the generalization bound holds.**
> To address this concern, we have clarified that Theorem 3.1 in the original paper (Theorem A.1 of the revised paper), only assumes that the labeled training set of size $m$ is randomly sampled uniformly from all the $N$ nodes of the graph without replacement, the MSE loss is bounded, and $m \ge cN$ for a constant $c$ where. Such assumptions are very mild and standard in the FSNC and graph learning/transductive learning literature. In particular, uniform sampling of the labeled training set without replacement is widely adopted in FSNC [3, 4, 5] and semi-supervised/transductive node classification in [6]. The assumptions about bounded MSE loss and  $m \ge cN$ are used in the generalization analysis of standard transductive learning in [7, 8].
> We have also clarified that our assumptions do not rely on homophily or community structure, which suggests that the generalization bound holds for both homophilic and heterophilic graphs. We have also demonstrated the effectiveness of LR-FGDM on both homophilic graph datasets, as shown in Table 1 and Table 6, and heterophilic graph datasets, as shown in Table 10 of the revised paper.
>
> We have also addressed the issues from Reviewer Kffe and Reviewer 678A in our rebuttals and the revised paper.

---

> ### Author Response · Authors · 2025-12-02
> **References in the Summary of Revision**
>
> **References**
>
> [1] Wang, Yancheng, Changyu Liu, and Yingzhen Yang. "Diffusion on Graph: Augmentation of Graph Structure for Node Classification." Transactions on Machine Learning Research (TMLR) 2025.
>
> [2] Casella, George, and Roger Berger. Statistical inference. Chapman and Hall/CRC, 2024.
>
> [3] Wu, Zongqian, et al. "Self-training based few-shot node classification by knowledge distillation." Proceedings of the AAAI conference on artificial intelligence. Vol. 38. No. 14. 2024.
>
> [4] Ding, Kaize, et al. "Meta propagation networks for graph few-shot semi-supervised learning." Proceedings of the AAAI conference on artificial intelligence. Vol. 36. No. 6. 2022.
>
> [5] Wu, Zongqian, et al. "Information Augmentation for Few-shot Node Classification." IJCAI. 2022.
>
> [6] Kipf, Thomas N., and Max Welling. "Semi-Supervised Classification with Graph Convolutional Network." ICLR 2017
>
> [7] Y. Yang. A New Concentration Inequality for Sampling Without Replacement and Its Application for Transductive Learning. ICML 2025.
>
> [8] Tolstikhin et al. Localized Complexities for Transductive Learning. COLT 2014.

---

### Meta-Review · Area_Chair_NWMg · 2026-01-08

**Summary:**

Reviewers noted the following concerns:

- Limited performance gains and whether the results are significant
- Lack of ablation study on the diffusion (LDM) component
- Lack of details on the scope/assumption of the main theorem
- Lack of details on certain technical designs and implementation details
- Comparison with DoG is potentially unfair

**Reviewer Concerns:**

The concerns above are largely addressed. The authors included by many additional experiments and clarifications that are deemed comprehensive and on-the-point.

**Reviewer Scores:**

Increase by 0.5-1 on average.

---

### Decision · Program_Chairs · 2026-01-26

Accept (Poster)